# The scaled-invariant Planckian metal and quantum criticality in $Ce_{1-x}Nd_xCoIn_5$

Yung-Yeh Chang[1,2], Hechang Lei [3,4], C. Petrovic [3] ✉ & Chung-Hou Chung [1,2] ✉

The mysterious Planckian metal state, showing perfect $T$-linear resistivity associated with universal scattering rate, $1/\tau = \alpha k_B T/\hbar$ with $\alpha \sim 1$, has been observed in the normal state of various strongly correlated superconductors close to a quantum critical point. However, its microscopic origin and link to quantum criticality remains an outstanding open problem. Here, we observe quantum-critical $T/B$-scaling of the Planckian metal state in resistivity and heat capacity of heavy-electron superconductor $Ce_{1-x}Nd_xCoIn_5$ in magnetic fields near the edge of antiferromagnetism at the critical doping $x_c \sim 0.03$. We present clear experimental evidences of Kondo hybridization being quantum critical at $x_c$. We provide a generic microscopic mechanism to qualitatively account for this quantum critical Planckian state within the quasi-two dimensional Kondo-Heisenberg lattice model near Kondo breakdown transition. We find $\alpha$ is a non-universal constant and depends inversely on the square of Kondo hybridization strength.

Metallic behavior that goes against the Landau's Fermi liquid paradigm for ordinary metals has commonly been observed in a wide variety of strongly interacting quantum materials and yet the emergence of such metals is poorly understood. This unconventional metallic or non-Fermi liquid (NFL) behavior often exists near a magnetic quantum phase transition, and shows "strange metal (SM)" phenomena with (quasi-)linear-in-temperature resistivity and singular logarithmic-in-temperature specific heat coefficient[1,2].

The Planckian metal state constitutes a particularly intriguing class of SM states and has been observed in the normal state of unconventional superconductors, including cuprate superconductors[3], iron pnictides and chalcogenides[4–8], organic[9,10] and heavy-fermion compounds[10–13], and twisted bilayer graphene[14]. The heart of this puzzling state is that the universal $T$-linear scattering rate reaches the Planckian dissipation limit allowed by quantum mechanics, $1/\tau = \alpha k_B T/\hbar$ with $\alpha \sim 1$[10,15–17]. This intriguing observation leads to fundamental questions: Is $\alpha$ a universal constant independent of microscopic details[18]? It was estimated that the Planckian bound can be reached in the low-temperature strong-coupling limit of electron-phonon and marginal

Fermi liquid systems[17]. In quantum critical systems, however, it is not clear if and how $\alpha$ depends on microscopic coupling constant at quantum critical point (QCP) and its link to the quantum critical scaling in observables. It was argued that in quantum critical systems with $T$-linear scattering rate, the Planckian time $\tau_{Pl} \equiv \hbar/k_B T$ is reached naturally ($\alpha \sim 1$) as a result of the scaled-invariant observables $F(\hbar\omega/k_B T) = F(\omega\tau_{Pl})$ at criticality[17]. However, such expectation has not yet been confirmed by a microscopic mechanism in any quantum critical systems.

While it is challenging to address these questions in cuprates, pnictide and organic superconductors, significant experimental and theoretical progress have been made in heavy-fermion quantum critical superconductors[19]. A prototypical example of such systems is the $CeMIn_5$ family with $M = Co$, Rh, Ir under field and pressure where superconductivity and non-Fermi liquid (NFL) normal state properties emerge due to competition between antiferromagnetism and Kondo correlation near a antiferromagnetic Kondo-breakdown (AF-KB) QCP[20–25]. In particular, the $T$-linear resistivity in $CeCoIn_5$ was reported to exhibit Planckian dissipation ($\alpha \sim 1$)[10], and therefore the Co-115 family is well suited for this study. In this work, we focus on Nd-doped $CeCoIn_5$[26].

[1]Physics Division, National Center for Theoretical Sciences, Taipei 10617 Taiwan, Republic of China. [2]Department of Electrophysics, National Yang-Ming Chiao-Tung University, Hsinchu 300 Taiwan, Republic of China. [3]Condensed Matter Physics and Materials Science Department, Brookhaven National Laboratory, Upton, NY 11973-5000, USA. [4]Present address: Department of Physics and Beijing Key Laboratory of Opto-electronic Functional Materials & Micro-nano Devices, Renmin University of China, Beijing 100872, People's Republic of China. ✉e-mail: petrovic@bnl.gov; chung0523@nycu.edu.tw

Single crystals $Ce_{1-x}Nd_xCoIn_5$ were grown and characterized in ref. [26]. Previous studies showed that antiferromagnetism coexists with superconductivity for $0.02 < x < 0.17$[26]. Here, we investigate further the Planckian state by applying a magnetic field in this material close to the edge of antiferromagnetism at $x \approx 0.02$. The electrical resistivity and heat capacity measurements were performed in a quantum design PPMS-9 system. The doping-field-temperature $(x, B, T)$ phase diagram of this material is shown in Fig. 1a. Doping Nd into the pure $CeCoIn_5$ effectively introduces chemical pressure and reduces Kondo coupling, therefore favors the long-range antiferromagnetism[25,26]. At zero magnetic field, pure $d$-wave superconducting ground state exists for very low Nd doping $0 < x \leq 0.02$[27], followed by a co-existing antiferromagnetic superconducting state for $0.05 \leq x < 0.17$, and the long-ranged antiferromagnetic phase is reached for $x > 0.17$[26]. For $0 < x < 0.17$, the resistivity at finite temperatures shows a maximum at temperature $T_{coh} \sim 45$K where coherent Kondo hybridization is reached, a generic feature of many heavy-fermion superconductors, it then drops to zero at $1K < T_c < 2K$ where superconductivity emerges[20,26].

The SM behavior with linear-in-$T$ resistivity was observed in the intermediate temperature range $T_c < T < 20K$: $\rho(T) = \rho_0 + A_1 T$ with $\rho_0$ being residual resistivity extrapolated to zero temperature and $A_1(\alpha)$ being the slope. Via quantum oscillation experiments in ref. [10], the scattering rate $1/\tau$ for $x = 0$ extracted via the Drude formula, $\rho = m^*/ne^2\tau$, from the resistivity data combined with the carrier concentration $n_i$, electron effective mass $m_i^*$ associated with the band $i = \alpha$, $\beta$-band in the $T$-linear resistivity region has been shown to reach the Planckian dissipation limit, $\alpha = \hbar/k_B T \tau = (e^2 \rho/k_B T)\sum_i n_i/m_i^* = A_1(e^2/k_B)\sum_i (n_i/m_i^*) \sim 1$. In Table 1, we find similar values of $\alpha \sim O(1)$ for $x = 0, 0.02, 0.05$ by a distinct set of quantum oscillation measurements in ref. [28] (detailed analysis in Supplementary Notes 2 and 3). Interestingly, as shown in Fig. 2, we find the inverse slope $1/A_1$, $T_c$ and $T_{coh}$ all show linear dependence on $x$ over $0 < x < 0.15$. In particular, $T_{coh}$, depending mostly on the Kondo scale, is well approximated by a perfect linear relation to the inverse slope $1/A_1$ as well as $1/\alpha$,

$$T_{coh}(x) \propto 1/A_1(x) \propto 1/\alpha(x), \qquad (1)$$

up to a non-universal constant shift. Eq. (1) above suggests that the Planckian coefficient $\alpha$ inversely depends on the strength of Kondo

hybridization in the Planckian SM region, as the second possible scenario mentioned above. To further investigate how $\alpha$ depends on Kondo hybridization near the critical doping $x_c$ and its relation to the possible QCP hidden inside the superconducting dome, we apply an external magnetic field and study signatures of quantum critical scaling in resistivity and specific heat.

As shown in Fig. 3a, b, for $x = 0.02$ and $x = 0.05$, the linear-in-$T$ resistivity shows almost the same slopes $(A_1)$ for $0 \leq B \leq 70$ kOe. This strongly indicates that the Kondo hybridization strength approximately remains at a constant quantum critical fixed point value, associated with a QCP for $0.02 < x_c < 0.05$ at zero field or a quantum critical line for the above field range (Fig. 1a). The scenario of the QCP associated with the critical Kondo hybridization is further supported by the following evidences: (i) quantum-critical $T/B$ scaling behavior in the linear-in-$T$ resistivity regime (Fig. 3c, d), (ii) the universal Planckian scaling of electron scattering rate $\hbar/\tau = \alpha\Gamma$ with $\Gamma = \sqrt{(k_B T)^2 + (l\mu_B B)^2}$ with $\alpha \sim O(1)$ (Fig. 3e, f), (iii) the $T/T_{LFL}$-scaling in specific heat coefficient $\gamma(T/T_{LFL}) \equiv C/T$ (Fig. 4a, b) with $T_{LFL}$ being the Fermi-liquid crossover scale, (iv) the $T/B$-power-law scaling in the normal state $\gamma$-coefficient: $\gamma \sim \Phi(T/B)$ where $\Phi(z) \sim z^{-m}$ is an universal scaling function with $m \approx 0.46$ for $x = 0.02$ (Fig. 4a, c) and $m \approx 0.5$ for $x = 0.05$ (Fig. 4b, d), and (v) power-law singular specific heat coefficient in the SM state: $C/T|_{T=0.1K} \sim |x - x_c|^{-\beta}$ with $\beta \approx 0.11$ for $x < x_c$ and $\beta \approx 0.16$ for $x > x_c$ (Fig. 1c). A more accurate estimation of the location of this hidden QCP reveals $x = x_c \approx 0.03$ by extrapolating the singular specific heat coefficient in the SM state under fields as Nd doping is tuned across the transition (Fig. 1c)[4,29]. When superconductivity is fully suppressed by magnetic field, a crossover from the non-Fermi liquid SM to the Fermi-liquid state with $T^2$ resistivity is clearly observed on two sides of the transition ($x = 0.02$ and $x = 0.05$) (Fig. 3a, b). This indicates a putative quantum critical (QC) line extended from the QCP hidden under superconducting dome at zero field and Nd doping (Fig. 1a)[25].

To qualitatively understand the above strange metal behaviors and more importantly the relation between Planckian coefficient $\alpha$ and Kondo hybridization near the QCP at $x_c$, here we propose a microscopic mechanism. It is based on the interplay between Kondo screening of a local $f$-electron ($f_{i\sigma}$ on site $i$ with spin $\sigma$), by mobile electrons ($c_{i\sigma}$), and the AF correlations between nearest-neighbor spins

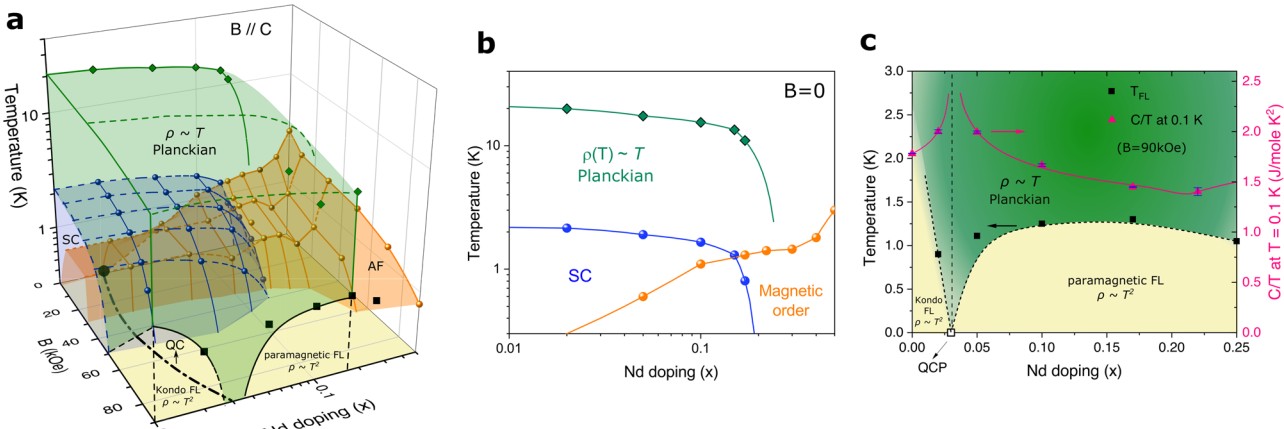

**Fig. 1 | The phase diagram for $Ce_{1-x}Nd_xCoIn_5$ for $B \| c$. a** The $(x, B, T)$ phase diagram: the antiferromagnetic (AF) Néel state occurs below the orange area and the superconducting (SC) phase sets in under the blue area (symbols). The data of zero-field plane is reproduced from Fig. 6 of ref. [26]. A QCP is predicted on the doping axis near $x \approx 0.03$ at zero field plane, hidden beneath the coexisting phase of the AF and SC phases. At temperature above the SC and AF phases and below the green region, this material shows linear-in-$T$ strange-metallic (SM) behavior. A quantum critical (QC) line (black dotted-dashed line) is expected to exist at finite field and Nd doping, connected to the QCP at zero field, $x = x_c$. **b** Phase diagram of **a** on the

zero-field facet. **c** Phase diagram of the system at $B = 90$ kOe. As the external field is large enough (approximately $B \sim 70$ kOe for $B\|c$), a QCP emerges at $x \approx 0.03$ as inferred from the $T^2$ resistivity measurement (below $T_{FL}$). This QCP is located at $B = 90$ kOe along the QC line of **a**. The enhancement of the normal-state electronic specific heat $C/T$ at the low-temperature limit ($T = 0.1$ K) is fitted by a power-law singular function at $x = x_c \approx 0.03$, $C/T|_{T=0.1K} \sim |x - x_c|^{-\beta}$, with $\beta \approx 0.11$ for $x < x_c$ and $\beta \approx 0.16$ for $x > x_c$ (pink curves, right axis). Data of **c** is measured at $B = 90$ kOe. The error bar of $C/T$ comes from the uncertainty of tuning temperature.

**Table 1 | The band-specific parameters of Ce$_{1-x}$Nd$_x$CoIn$_5$**

| | $x = 0$ $(\theta = 0)$ | $x = 0.02$ $(\theta = 3^\circ)$ | $x = 0.05$ $(\theta = 2^\circ)$ | $x = 0.1$ $(\theta = 7^\circ)$ |
|---|---|---|---|---|
| $F$(kT) ($\alpha$ band) | $4.9 \pm 0.5$ | $4.89 \pm 1.02$ | $4.88 \pm 0.5$ | $4.41 \pm 0.41$ |
| $n$ ($\times 10^{28}$ m$^{-3}$) ($\alpha$ band) | $0.31 \pm 0.04$ | $0.32 \pm 0.07$ | $0.26 \pm 0.02$ | $0.17 \pm 0.02$ |
| $m^*$($m_0$) ($\alpha$ band) | $11.7 \pm 2.6$ | $11.7 \pm 4.3$ | $9.15 \pm 2.0$ | $7 \pm 1.0$ |
| $A_1$($\mu\Omega \cdot$ cm/K) | $0.98 \pm 0.07$ | $1.0 \pm 0.07$ | $1.17 \pm 0.08$ | $1.49 \pm 0.1$ |
| $\alpha$ ($\alpha$ band + $\beta$ band) | $0.7 \pm 0.2$ | $0.72 \pm 0.4$ | $0.85 \pm 0.2$ | $0.96 \pm 0.2$ |

The averaged dHvA frequencies $F$, carrier concentration $n$, and effective masses $m^*$ of the $\alpha$ band shown here are taken from ref. [28]. Carrier concentration $n$ of the $\alpha$-band for $x = 0.05$ is estimated by a $d$-dimensional Fermi volume with $d = 2.45$, while it is estimated by a two-dimensional Fermi volume for $x = 0$, $0.02$, in accordance with the angular dependence of the dHvA frequencies measured in ref. [26]. Here, $m_0$ denotes the bare electron mass, while $\alpha$ and $A_1$ are extracted from the linear-in-temperature resistivity at zero field from ref. [26]. The error bars of $F$, $n$, and $m^*$ represent the standard deviation. The error bars of $\alpha$ and $A_1$ shown here are the same as described in Fig. 2. Note that the $\alpha$ coefficient shown here contains a sub-leading contribution from the $\beta$ band, estimated by the band parameters for pure CeCoIn$_5$ in Ref. [10]: $n \approx 0.63 \times 10^{28}$ m$^{-3}$ (corresponding to the average dHvA frequency $F \approx 9.75$ kT) and $m^* \approx 100 m_0$ (Supplementary Notes 2 and 3 for details).

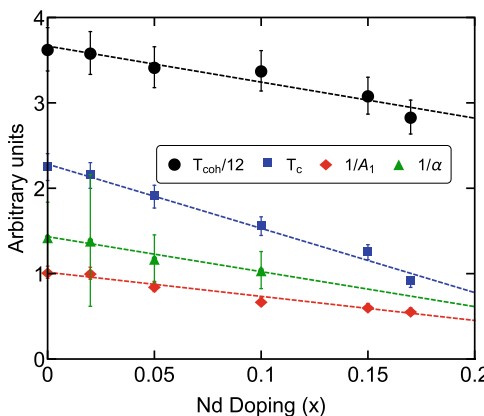

**Fig. 2 | Linear-in-$x$ dependence of $T_{coh}$, $T_c$ and $1/A_1$, and $1/\alpha$.** The data are taken from ref. [26] and Table 1. The error bars of $T_{coh}$, $T_c$ and $1/A_1$ arise from the inaccuracy of tuning temperature, while the error bar for $1/\alpha$ includes the uncertainties of $m^*$, $n$, and $F$ shown in Table 1.

of local $f$-electrons in the form of quasi-2$d$ ($d = 2 + \eta$ with $0 < \eta < 1$) Kondo-Heisenberg (KH) model (see "Methods" section). This mechanism has been used to qualitatively describe the AF QCP at $x_c$ inside the superconducting dome of CeRhIn$_5$ under pressure[30,31], which bears a striking similarity to that for Ce$_{1-x}$Nd$_x$CoIn$_5$ at zero field under Nd doping. The Anderson's resonating-valence-bond (RVB) spin-liquid state[32], defined by the spatially homogeneous bosonic spin-singlet pair, $\Delta_{RVB} \equiv J_H \sum_\sigma \langle \text{sgn}(\sigma) f_{i\sigma}^\dagger f_{j,-\sigma}^\dagger \rangle$, on nearest-neighbor sites $i$, $j$, is introduced here for the AF Heisenberg term with the exchange coupling $J_H$. The Kondo hybridization is described by the spatially homogeneous bosonic $\chi$ field where, at the mean-field level, $\chi \equiv J_K \sum_\sigma \langle f_{i\sigma}^\dagger c_{i\sigma} \rangle$ with Kondo coupling $J_K$. At the edge of antiferromagnetism, the Kondo effect not only stablizes the RVB spin liquid against the magnetic long-ranged order by partially sharing the $f$-electron spins[33], but also introduces hoping of the RVB bonds to the conduction band to form charged Cooper pairs. This leads to a Kondo-RVB coexisting heavy-electron superconducting state with estimated transition temperature $T_c \sim \chi^2 \Delta_{RVB}$, in quantitatively good agreement with the experimental observation[34].

Via the competition between Kondo and RVB physics, an AF-KB QCP was predicted inside the superconducting dome, qualitatively describing the phase transition between AF-superconducting coexisting phase and a pure superconducting phase observed in CeRhIn$_5$[34], similar to our case here. By analyzing the amplitude fluctuations of the Kondo and RVB correlations beyond mean-field level via renormalization group (RG) analysis[34,35], this mechanism provides a qualitative and semi-quantitative understanding of the SM properties in CeCoIn$_5$ in fields and pressure near the AF-KB QCP[20,24,25]. In particular, it captures the observed $T$-linear resistivity in terms of critical Kondo (charge)

fluctuations ($\hat{\chi}$ field) via the electron-phonon-like interaction, $\hat{H}_K \sim J_K \sum_{i\sigma} (c_{i\sigma}^\dagger f_{i\sigma} \hat{\chi}_i + H.c.)$ (see "Methods" section), and a power-law-in-$T$ divergence in $\gamma(T)$ via both Kondo and RVB fluctuations[34,35]. The quasi-2$d$ nature of our theoretical framework, essential to the KB transition, is consistent with the quasi-2$d$ nature of the Fermi surface in CeCoIn$_5$[36] as well as the dimensional crossover in Fermi surface evolution of the $\alpha$-band from 2$d$-like to 3$d$-like in our system[28] with increasing Nd concentrations observed in quantum oscillations. By analyzing the data of quantum oscillation in ref. [28], we observed a sudden change in charge carrier $n$ across $x_c$ as $T \to 0$ (Table 1), a possible signature of a jump in Fermi surface volume at ground state. This signature is consistent with our KB QCP scenario. Interestingly, evidence of a delocalization (KB) transition has been observed in a closely related compound CeCo(In$_{1-x}$Sn$_x$)$_5$ near $x \sim 0.016$[37]. It is promising to expect that the KB QCP might also occur here.

We now apply the above theoretical framework to investigate how the Planckian coefficient $\alpha$ depends on the Kondo hybridization. First, the coherent Kondo temperature $T_{coh}$ is related to the condensation amplitude of Kondo hybridization $\chi$ through $T_{coh} \sim \chi^2/D$[34,38]. Since $\chi$ is proportional to the Kondo coupling $J_K$, this suggests $T_{coh} \sim J_K^2/D$. From Eq. (1), we have

$$\alpha(x) \propto A_1(x) \propto 1/T_{coh}(x) \propto 1/J_K^2 \tag{2}$$

up to a non-universal constant shift. Meanwhile, within our theoretical framework, the superconducting transition temperature $T_c$ of an antiferromagnetic quantum critical heavy-fermion superconductor is related to the RVB and Kondo correlations via[34,39]

$$T_c \propto \chi^2 \Delta_{RVB} \propto J_K^2 \Delta_{RVB}. \tag{3}$$

The experimentally observed linear relation (up to a non-universal constant) between $T_c$ and the inverse slope $1/A_1$: $T_c(x) \propto 1/A_1(x) \propto 1/\alpha(x)$ and Eq. (3) implies that $\alpha \propto 1/J_K^2$, consistent with Eq. (2) above.

The approximated constant value of $\alpha \sim O(1)$ seen for $0 \le x \le 0.05$ (with an error bar, Table 1) can therefore be understood as the Kondo hybridization being close to quantum critical. This is also in good agreement with the same slope observed in linear-in-$T$ resistivity near the edge of antiferromagnetism as well as the decreasing $1/A_1$ (or Kondo correlation) with increasing Nd doping (Fig. 2). Furthermore, the experimentally and theoretically supported relations between $A_1$ and the Kondo coupling $J_K$, $\alpha \propto A_1 \propto 1/J_K^2$ strengthen the link between the Planckian $T$-linear scattering rate and critical Kondo hybridization in our system.

A few remarks are made before we conclude. Firstly, in the SM region, the electron effective mass $m^*$ in general may be temperature dependent. In such case, the Planckian scattering rate is reached only near the lower end of the linear-in-$T$ resistivity region where $m^*$ is well-approximated by a temperature-independent constant extracted from quantum oscillation measurement in the Fermi-liquid region[17].

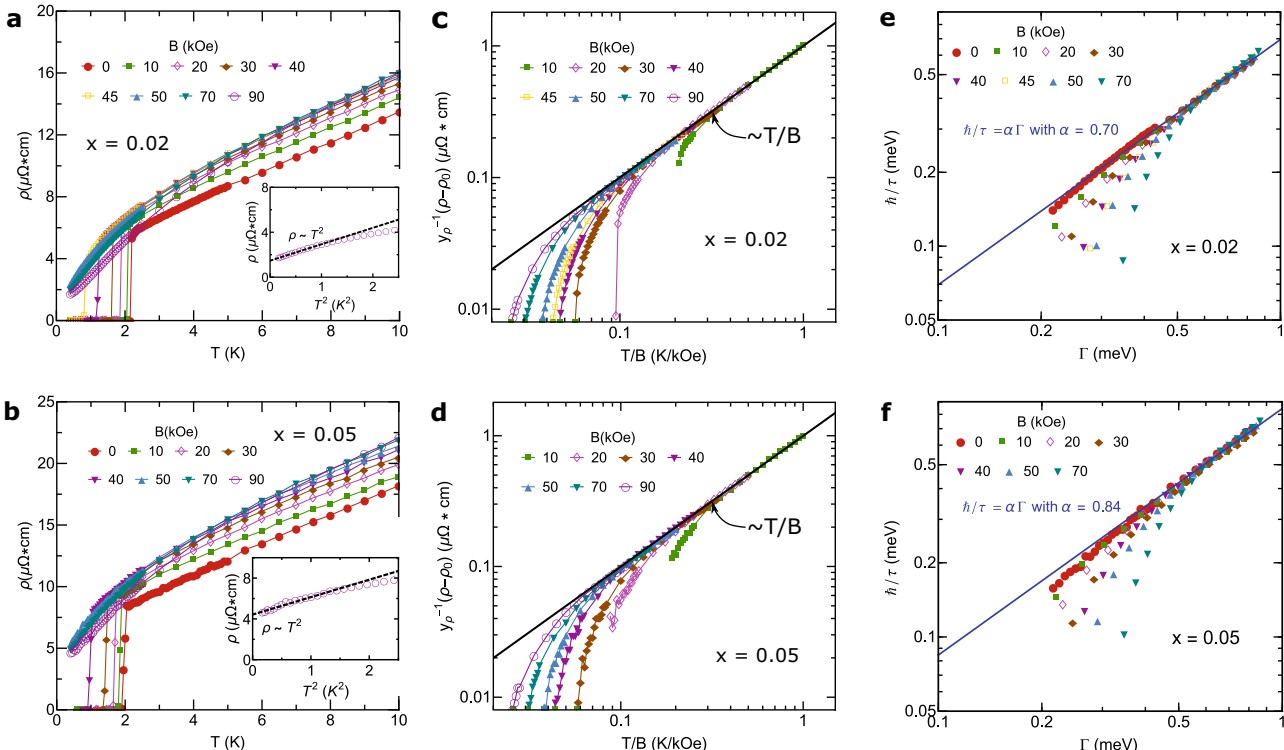

**Fig. 3 | Electrical resistivity and the scaling. a, b** The resistivity $\rho(T)$ of Ce$_{1-x}$Nd$_x$CoIn$_5$ with $x = 0.02$ and $x = 0.05$, respectively, for $B \| c$. The insets of **a, b** show the Fermi-liquid behavior with $\rho(T)$ - $T^2$ electrical resistivity at low temperatures for $B = 90$ kOe. **c, d** The $T/B$ scaling of both **a** and **b** show an universal function linearly proportional to $T/B$ (black solid lines). $y_\rho$ is a non-universal (field-dependent) constant, satisfying $y_\rho^{-1}[\rho(T) - \rho_0] = T/B$. **e, f** The scattering rates $\hbar/\tau$ for $x = 0.02$, 0.05 at the intermediate temperature regime where the electrical resistivity exhibits a linear-in-$T$ dependence show an universal scaling function $\hbar/\tau = \alpha\Gamma$ (blue solid lines), with $\Gamma \equiv \sqrt{(k_B T)^2 + (l\mu_B B)^2}$, $\tau$ being the relaxation time, $\mu_B$ being the Bohr magneton, and $l = 0.67$ being a fitting parameter[43]. In **e** and **f**, the $\beta$ band of pure CeCoIn$_5$, with the same band parameters used in Table 1, is taken into account in performing the $\Gamma$ scaling of resistivity. The extracted $\alpha$ coefficients shown in **e** and **f** are consistent with that shown in Table 1.

Nevertheless, we have checked within our theoretical framework that the temperature dependence of $m^*$ in our SM region is negligible. As a result, the Planckian state in our system extends over the entire SM region. Secondly, though the Drude formula has been widely used to relate the transport and scattering rate in the Fermi-liquid region and near the lower end of the $T$-linear region, whether it is valid deep in the non-Fermi liquid SM region is an open question[17,40]. Thirdly, while quantum oscillation measurement is widely used to estimate $n$ and $m^*$, different approach by Hall coefficient and $A$ coefficient associated with the Fermi-liquid behavior with $T^2$ resistivity has been used[40]. Finally, the above quantum critical features in specific heat coefficient are inconsistent with the conventional spin-density-wave (SDW) theory[41,42] for the AF QCP though the SDW fluctuations are expected to appear.

In summary, we observe the quantum-critical $T/B$-scaling of the Planckian metal state in the resistivity and specific heat coefficient of heavy-electron superconductor Ce$_{1-x}$Nd$_x$CoIn$_5$ under fields. Clear experimental evidences are shown to support the notion of Kondo hybridization being quantum critical at $x_c \sim 0.03$. We propose a microscopic mechanism based on the quasi-2$d$ Kondo-Heisenberg lattice model to qualitatively account for the observed strange metal behaviors. Furthermore, we find that the Planckian coefficient $\alpha$ for Nd-doped CeCoIn$_5$ shows an inversely quadratic dependence in the Kondo coupling, as inferred from our theoretical framework as well as the linear relation between $1/\alpha$, the inverse slope of $T$-linear resistivity $1/A_1$ and the coherence temperature $T_{\mathrm{coh}}$. Our observation and proposed mechanism offer the first microscopic understanding of the Planckian dissipation limit in a quantum critical system. This generic mechanism is relevant for other heavy-fermion quantum critical superconductors showing Planckian metal states, such as CeRhIn$_5$ and related compounds. Our results motivate further experimental study, such as the Hall coefficient measurement on the signatures of Kondo breakdown near the critical Nd doping as well as theoretical studies, such as more accurate predictions on the value of $\alpha$ and critical exponents in scaling behaviors of observables, and a fundamental issue on whether quantum critical systems with $T$-linear resistivity, in general, implies the Planckain metals.

## Methods
### A microscopic model
Our start point is the microscopic large-$N$ (Sp($N$)) Kondo-Heisenberg Hamiltonian $H = H_0 + H_K + H_J$, where

$$H_0 = \sum_{\langle i,j \rangle;\sigma} \left[ t_{ij} c_{i\sigma}^\dagger c_{j\sigma} + H.c. \right] - \sum_{i\sigma} \mu c_{i\sigma}^\dagger c_{i\sigma},$$
$$H_J = \sum_{\langle i,j \rangle} \frac{J_{ij}}{N} \mathbf{S}_i^f \cdot \mathbf{S}_j^f, \quad H_K = \frac{J_K}{N} \sum_i \mathbf{S}_i^f \cdot \mathbf{s}^c. \quad (4)$$

In Eq. (4), $H_0$ describes hopping of conduction ($c_{i\sigma}$) electrons. The antiferromagnetic Heisenberg interaction between two impurity-spin ($\mathbf{S}^f$) of the electrons occupying the localized $f$ orbitals of neighboring sites is described by $H_J$ and the Kondo screening of impurity spins by conduction electrons is captured by $H_K$. Via Hubbard-Stratonovich transformation, $H_K$ and $H_J$ can be factorized as

$$H_J \rightarrow \sum_{\langle i,j \rangle;\alpha,\beta} \left[ \Phi_{ij} \mathscr{J}^{\alpha\beta} f_{i\alpha} f_{j\beta} + H.c. \right] + \sum_{\langle i,j \rangle} N \frac{|\Phi_{ij}|^2}{J_H},$$
$$H_K \rightarrow \frac{1}{\sqrt{N}} \sum_{i,\sigma} \left[ \left( c_{i\sigma}^\dagger f_{i\sigma} \right) \chi_i + H.c. \right] + \sum_i \frac{|\chi_i|^2}{J_K}. \quad (5)$$

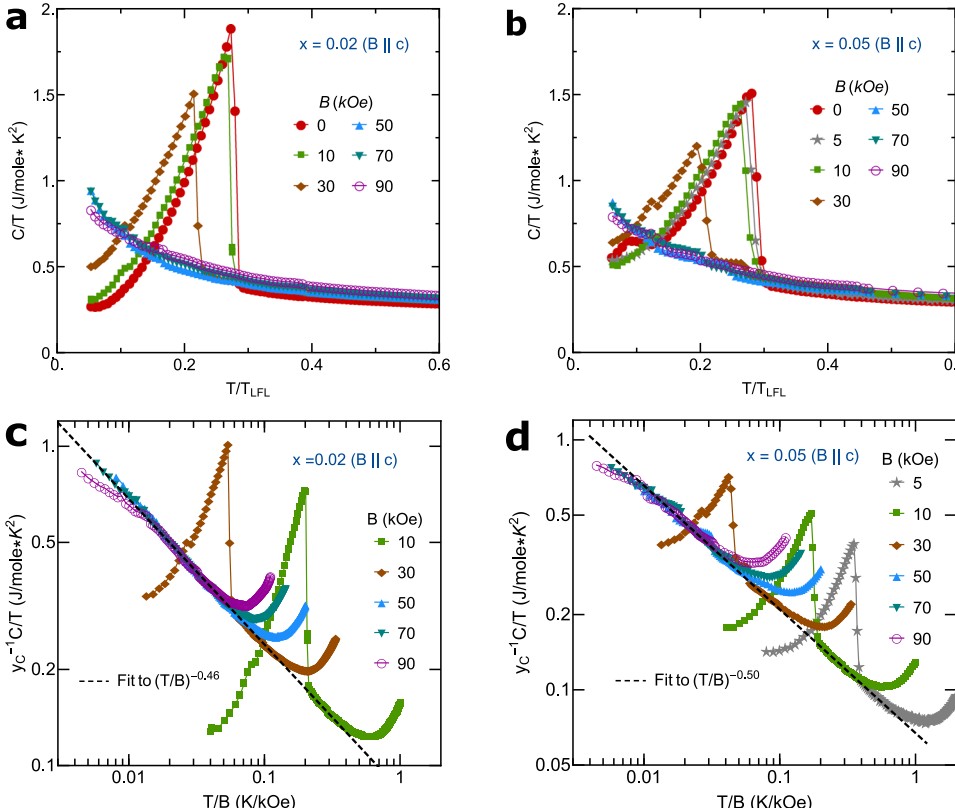

**Fig. 4 | Specific heat coefficient $C/T$ and its $T/B$ scaling. a, b** show the electronic specific heat coefficient $C/T$ versus $T/T_{LFL}$ with different fields $B\|c$ for 0.02, and 0.05, respectively while **c** and **d** display the power-law $T/B$ scaling of **a** and **b**. Here, $y_C$ in **c** and **d** represents a non-universal scaling parameter, obeying $y_C^{-1}(C/T) = \Phi(T/B)$ with $\Phi$ being a power-law function here.

Here, the local spin operator $\mathbf{S}_i^f$ is represented in terms of the constraint fermionic Sp($N$) fields, $f_{i\alpha}$, which needs to be subjected to the constraint, $\langle \sum_\sigma f_{i\sigma}^\dagger f_{i\sigma} \rangle = N\kappa$, to ensure its local nature. The constant $\kappa < 1$ allows us to capture the valence fluctuations. Thus, to capture the local constraint, an additional term has to be included in the Hamiltonian, $H_\lambda = \sum_{i,\sigma} \lambda \left[ f_{i\sigma}^\dagger f_{i\sigma} - Q \right]$ with $\lambda$ being the Lagrange multiplier. In the $H_J$ term, we assume an uniform antiferromagnetic RKKY coupling $J_{ij} = J_H$ on a lattice where $i, j$ are nearest-neighbor site indices, $\mathscr{J}^{\alpha\beta} = \mathscr{J}_{\alpha\beta} = -\mathscr{J}^{\beta\alpha}$ denotes the anti-symmetric tensor, $\sigma, \alpha, \beta \in \{-\frac{N}{2}, \cdots, \frac{N}{2}\}$ label the spin indices. Here, $\chi_i$ and $\Phi_{ij}$ are the spatially dependent Hubbard-Stratonovich fields, where their (spatially uniform) mean-field values, $\chi \equiv \langle \frac{J_K}{\sqrt{N}} \sum_\sigma f_{i\sigma}^\dagger c_{i\sigma} \rangle$ and $\Delta_{RVB} \equiv \langle \Phi_{ij} \rangle = \langle \frac{J_H}{N} \sum_{\alpha,\beta} \mathscr{J}_{\alpha\beta} f_i^{\alpha\dagger} f_j^{\beta\dagger} \rangle$, can be used as order parameters to describe the Kondo-screened heavy Fermi liquid state and the RVB spin-liquid state.

While considering the fluctuations of $\chi_i$ and $\Phi_{ij}$ beyond mean-field level, we can express $\chi_i \rightarrow \chi + J_K \hat{\chi}_i$ and $\Phi_{ij} \rightarrow \Delta_{RVB} + J_H \hat{\Phi}_{ij}$, where $\hat{\chi}_i$ and $\hat{\Phi}_{ij}$ denote the fluctuation of $\chi$ and $\Delta_{RVB}$. Both $H_K$ and $H_J$ can be divided into the mean-field and fluctuating parts, namely $H_K \rightarrow H_K^{MF} + \hat{H}_K$ and $H_J \rightarrow H_J^{MF} + \hat{H}_J$ with $H_J^{MF}$ and $H_K^{MF}$ at mean-field, and $\hat{H}_J$ and $\hat{H}_K$ beyond mean-field level. The microscopic model relevant for the heavy-fermion materials therefore becomes $H \rightarrow H_0 + H_\lambda + H_J^{MF} + H_K^{MF} + \hat{H}_K + \hat{H}_J$.

## Data availability

Original data and codes created for the study have been deposited in the online Zenodo repository (https://doi.org/10.5281/zenodo.7187946).

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

## Acknowledgements

We acknowledge discussions with S. Kirchner, J. Thompson, A. P. Mackenzie, and C.-L. Huang. This work is supported by the Ministry of Science and Technology Grants 107-2112-M-009-010-MY3, the National Center for Theoretical Sciences of Taiwan, Republic of China (C.H.C.). Work at BNL is supported by the Office of Basic Energy Sciences, Materials Sciences and Engineering Division, U.S. Department of Energy (DOE) under Contract No. DESC0012704 (materials synthesis, thermodynamic and transport characterization).

## Author contributions

C.H.C. conceived the study and led the project. The crystals were synthesized by H.C.L. and C.P. Measurements of electrical resistivity and specific heat were performed by H.C.L. and C.P. The experimental data were analyzed by Y.Y.C., C.H.C., C.P., and H.C.L..Theoretical calculations were performed by Y.Y.C. and C.H.C. The manuscript was written by Y.Y.C., C.H.C., and C.P. All authors participated in discussions.

## Competing interests

The authors declare no competing interests.
