## [Peer Review File · Nature Communications]

REVIEWER COMMENTS

Reviewer #1 (Remarks to the Author):

Chang et al. has presented a combination of experimental and theoretical work in the current manuscript to clarify the nature of Planckian scattering of carriers in the model heavy-fermion compound CdNdCoIn in the quantum critical regime. Whether Planckian scattering universally applies to a wide range of strange metals is a timely topic. The methodology adopted by the authors to perform the present analysis would be interesting to be extended to other famous compounds in this context. On the experimental side, they performed transport and specific heat measurements in the doping-temperature-field phase diagram of CdNdCoIn and established a few observations: (1) The slope of linear-in-T resistivity (which is proportional to the Planckian α coefficient) is inversely proportional to the Kondo hybridization strength. (2) Dynamical scaling of resistivity, scattering rate, and specific heat suggests quantum criticality. Theoretically, a Kondo-Heisenberg model was developed, establishing the connection between the α coefficient within the expression for the scattering rate and the Kondo hybridization in the critical regime. The analysis clarifies the microscopic mechanism for Planckian scattering observed in this compound. It therefore unifies the physics of a few concepts, including strange metal behavior, Planckian scattering, and AF-KB quantum criticality. While I view this work as an interesting contribution to the field, a few issues need to be addressed.

1. In Fig 3c and d showing T/B scaling of resistivity, what is the prefactor of γ_{ρ}^{-1} ? I don't find an interpretation of this quantity and how it is chosen for each trace to perform the scaling analysis. If the equation of scattering rate in line 109 is true ($1/\tau$ proportional to $\sqrt{T^2 + B^2}$), wouldn't this prefactor simply be the temperature itself? The same question holds for the " γ_c^{-1} " in the specific heat scaling graphs.

2. It was emphasized that the α coefficient is in general dependent on Kondo hybridization, and it's only at temperatures very close to the Fermi liquid phase can the Kondo coupling coefficient be reduced to the effective dimension, making α approach 1 (approaching Planckian scattering). This is shown by the narrow blue shaded region in Fig. 1b. I wonder why the Planckian scattering regime is so narrow along the temperature axis. The linear-in-T resistivity regime is much wider, and as long as ρ is linear in T, α would be a constant across the same T, so the entire T range might behave as "Planckian" as the low-T region. This question also applies to the author's comment about T-dependence of carrier mass around line 257. The statement is true for many compounds, but if m^* is so T-dependent in the strange metal T range, why ρ would be linear in T in the first place?

3. Can the author's model capture the B-field dependence of resistivity/specific heat? The current model captures T dependence but I wonder if T/B scaling can emerge from it also.

Minor aspects:

1. The phase diagram in Fig. 1 needs some improvements. The B=0 side of the diagram is not clear visually. I suggest adding a panel c to show the projection of the 3D phase diagram onto the B=0 facet.

2. On line 123, the reference to Fig. 3c seems to be problematic.

Reviewer #2 (Remarks to the Author):

This paper tried to get the quantum-critical T/B-scaling of the resistivity and heat capacity data of heavy-electron superconductor $\text{Ce}_{1-x}\text{Nd}_x\text{CoIn}_5$. Then the authors provide a microscopic mechanism to account for the Planckian state in a quantum critical system based on the critical charge fluctuations near Kondo breakdown transition at x_c within the quasi-two-dimensional Kondo-Heisenberg lattice model.

I find this paper is not well organized and it is hard for the readers to follow the authors to get a convincing conclusion. I don't think this paper is suitable for Nature Communications.

Some minor comments:

(1) In the caption of Table 1, $\text{Nd}_x\text{Ce}_{1-x}\text{CoIn}_5$ should be $\text{Ce}_{1-x}\text{Nd}_x\text{CoIn}_5$, in order to be consistent with other parts of this manuscript.

(2) The format of chemical formula in many references is not correct. For examples, $\text{baf}_2(\text{As}_{1-x}\text{P}_x)_2$ in Ref. 6, $\text{Fese}_{1-x}\text{S}_x$ in Ref. 8, cecoin_5 in Ref. 27, 38, 40.

Reviewer #3 (Remarks to the Author):

This paper contains some very interesting experimental results on strange metallic behavior with T-linear resistivity and Planckian dissipation in Nd-doped CeCoIn₅. Not only is T-linear resistivity found close to the critical doping, but other signatures of quantum criticality such as strongly interaction-renormalized effective electron mass, and universal magnetic field dependencies of the transport scattering rate are also found. The experimental observations by themselves seem enough to warrant publication of this paper in Nature Communications as far as I am concerned.

However, the theoretical analysis is hard to follow and not clearly presented, and I am therefore not able to determine its validity. I feel that it would be more appropriate to move the theoretical analysis to a separate paper in a specialized journal. I also am confused about certain steps in the theoretical analysis as detailed below:

1. How is the fermion self energy Σ_c , used in section S. V, computed? It would be useful to show the derivation of this. I do not understand how Eq. (S34) is derived!

2. Why is the fermion self energy being taken to be equal to the transport scattering rate (Eq. (S33))? What about vertex corrections contributing to transport? If the self-energy represents largely forward-scattering processes at the QCP, then it is NOT equal to the transport scattering rate. Again, whether this is true or not is not clear because of the highly compressed theoretical section of the paper.

3. Similarly, I do not understand the computations of section S. VI. What is the value of $\varepsilon_{\chi}(k)$? The derivation of this from the χ self energy should be provided!

For these reasons, I believe that the experimental part of the paper is very interesting and publishable, while the theoretical part should be moved to a separate paper in a specialized journal, and written up in a manner that provides systematic and clear derivations of all the necessary steps. If the authors can do this, I can recommend publication of their (experimental) work in Nature Communications.

Point-by-point reply to reviewer reports on “The scaled-invariant Planckian metal and quantum criticality in $\text{Ce}_{1-x}\text{Nd}_x\text{CoIn}_5$ ”

by Yung-Yeh Chang, Hechang Lei, C. Petrovic, and Chung-Hou Chung

October 12, 2022

We thank all of the three reviewers for their comments on our manuscript! In particular, we thank Reviewer 1 for the following supportive comments: “Whether Planckian scattering universally applies to a wide range of strange metals is a timely topic. The methodology adopted by the authors to perform the present analysis would be interesting to be extended to other famous compounds in this context. The analysis clarifies the microscopic mechanism for Planckian scattering observed in this compound. It therefore unifies the physics of a few concepts, including strange metal behavior, Planckian scattering, and AF-KB quantum criticality. While I view this work as an interesting contribution to the field, a few issues need to be addressed.”. We also thank the Reviewer 3 for the supportive comments: “This paper contains some very interesting experimental results on strange metallic behavior with T -linear resistivity and Planckian dissipation in Nd-doped CeCoIn_5 . Not only is T -linear resistivity found close to the critical doping, but other signatures of quantum criticality such as strongly interaction-renormalized effective electron mass, and universal magnetic field dependencies of the transport scattering rate are also found. The experimental observations by themselves seem enough to warrant publication of this paper in Nature Communications as far as I am concerned.” The merits of our paper and the significant contributions of our results to the corresponding research field are clearly appreciated in the above comments. Below, we present our point-by-point response to address all the questions, comments, and issues raised by the three reviewers in details.

Reviewer 1

Chang et al. has presented a combination of experimental and theoretical work in the current manuscript to clarify the nature of Planckian scattering of carriers in the model heavy-fermion compound CdNdCoIn in the quantum critical regime. Whether Planckian scattering universally applies to a wide range of strange metals is a timely topic. The

methodology adopted by the authors to perform the present analysis would be interesting to be extended to other famous compounds in this context. On the experimental side, they performed transport and specific heat measurements in the doping–temperature–field phase diagram of CdNdCoIn and established a few observations: (1) The slope of linear–in– T resistivity (which is proportional to the Planckian α coefficient) is inversely proportional to the Kondo hybridization strength. (2) Dynamical scaling of resistivity, scattering rate, and specific heat suggests quantum criticality. Theoretically, a Kondo–Heisenberg model was developed, establishing the connection between the α coefficient within the expression for the scattering rate and the Kondo hybridization in the critical regime. The analysis clarifies the microscopic mechanism for Planckian scattering observed in this compound. It therefore unifies the physics of a few concepts, including strange metal behavior, Planckian scattering, and AF–KB quantum criticality. While I view this work as an interesting contribution to the field, a few issues need to be addressed.

R1–A

In Fig 3c and d showing T/B scaling of resistivity, what is the prefactor of y_ρ^{-1} ? I don't find an interpretation of this quantity and how it is chosen for each trace to perform the scaling analysis. If the equation of scattering rate in line 109 is true ($1/\tau$ proportional to $\sqrt{T^2 + B^2}$), wouldn't this prefactor simply be the temperature itself? The same question holds for the " y_c^{-1} " in the specific heat scaling graphs.

R1–A–R

We thank Reviewer 1 for pointing out this issue. We agree with Reviewer 1 that y_ρ and y_C appear in Figures 3c, 3d, 4c and 4d in our previous draft was not properly described. A clear interpretation them is thus needed.

In this manuscript, y_ρ and y_C are both non–universal (field–dependent) scaling parameters we used when we perform the T/B –scaling for the linear–in–temperature (T) resistivity and the specific heat coefficient C/T in different magnetic fields, respectively. We describe below how we perform the T/B –scaling. First, we describe how we use y_ρ in the T/B –scaling for resistivity. The linear–in–temperature (T) resistivity data satisfies $\Delta\rho(T) \equiv \rho(T) - \rho_0 = A_1 T$, where ρ_0 denotes the residual resistivity at $T = 0$ and A_1 the slope of **T –linear resistivity**: Since the linear–in– T data of resistivity follows $\Delta\rho(T) = A_1 T$, we shall use it for our scaling analysis. We divide both sides by B , yielding

$$\frac{\Delta\rho(T)}{B} = A_1 \times \frac{T}{B}. \quad (1)$$

Dividing A_1 on both sides of Eq. (1), all resistivity data with different magnetic fields can collapse onto the curve, given by

$$y_\rho^{-1} \Delta\rho(T) = \frac{T}{B}. \quad (2)$$

with $y_\rho \equiv A_1 B$. The scaling function shown above also answers another question raised by Reviewer 1 in R1–A: he/she suspects that y_ρ is simply the temperature itself as inferred from the scaling behavior of scattering rate $\hbar/\tau \sim \sqrt{(k_B T)^2 + (l\mu_B B)^2}$. However, it is clear from Eq. (2) above that $y_\rho = A_1 B$, is different.

Similar scaling analysis for resistivity is applied to the scaling of specific heat coefficient C/T . At temperatures slightly above the superconducting transition temperature ($T > T_c$), the C/T data for different magnetic fields exhibits a power-law behavior such as

$$\frac{C}{T} = aT^{-\alpha} \quad (3)$$

with a being a non-universal (field-dependent) factor and α being field-independent. Here, we collapse all C/T data for magnetic fields $B < 90\text{kOe}$ onto the C/T curve with $B = 90\text{kOe}$. Below, we demonstrate the scaling procedures of C/T by directly exploiting Eq. (3): We define a_0 the a -factor at $B = 90\text{kOe}$. We first multiply a_0/a on both sides of Eq. (3), leading to

$$\frac{a_0}{a} \times \frac{C}{T} = a_0 T^{-\alpha}.$$

After dividing both sides of Eq. (3) by $B^{-\alpha}$, all C/T data with different magnetic fields can collapse onto a single power-law scaling function in terms of T/B , given by

$$y_C^{-1} \frac{C}{T} = a_0 \left(\frac{T}{B} \right)^{-\alpha} \quad (4)$$

with $y_C = \frac{1}{(a/a_0)B^{-\alpha}}$.

R1–A–C

To address the question in R1–A, we have clearly defined y_ρ and y_C in the revised manuscript: We added the sentence “ y_ρ is a non-universal (field-dependent) constant, satisfying $y_\rho^{-1}[\rho(T) - \rho_0] = T/B$.” to the caption of Figure 3 and “Here, y_C in c, obeying $y_C^{-1}(C_V/T) = \Phi(T/B)$ with Φ being a power-law function here.” to the caption of Figure 4.

R1–B

It was emphasized that the α coefficient is in general dependent on Kondo hybridization, and it's only at temperatures very close to the Fermi liquid phase can the Kondo coupling coefficient be reduced to the effective dimension, making α approach 1 (approaching Planckian scattering). This is shown by the narrow blue shaded region in Fig. 1b. I wonder why the Planckian scattering regime is so narrow along the temperature axis. The linear-in- T resistivity regime is much wider, and as long as ρ is linear in T , α would be a constant across the same T , so the entire T range might behave as “Planckian” as the low- T region. This question also applies to the author's comment about T -dependence of carrier mass around line 257. The statement is true for many compounds, but if m^* is so T -dependent in the strange metal T range, why ρ would be linear in T in the first place?

R1–B–R

In R1–B, Reviewer 1 wonders why the Planckian scattering regime is so narrow along the T axis in Figure 1b of the previous manuscript. He/She emphasizes that, since the linear–in– T resistivity regime is much wider, the Planckian regime should cover the the whole temperature range at which the resistivity is linear–in– T . Moreover, Reviewer 1 asked “if m^* is so T –dependent in the strange metal T range, why ρ would be linear–in– T in the first place?”

First, we would like to clarify the relation among the electron effective mass m^* , the linear–in– T resistivity and the Planckian scattering rate. A linear–in– T resistivity does not necessarily indicate the Planckian scattering rate. The Planckian scattering rate is defined as the T –linear scattering rate $1/\tau$ such that

$$\frac{1}{\tau} = \alpha \left(\frac{k_B T}{\hbar} \right) \quad (5)$$

with the proportionality constant α (the Planckian coefficient) being close to 1, i.e. $\alpha \approx 1$. The Planckian scattering rate may imply the linear–in– T resistivity when the carrier density and electron effective mass are both temperature independent. This result can be derived from the Drude’s formula: the electrical resistivity is given by

$$\rho = \frac{1}{e^2} \frac{m^*}{n} \frac{1}{\tau} \quad \text{or} \quad \frac{1}{\tau} = \frac{ne^2}{m^*} \rho \quad (6)$$

with n being the carrier density. Note, however, that it is the electrical resistivity that is directly measured in the experiments, and it shows T –linear behavior (regardless whether n, m^* are temperature dependent or not in the strange metal region). The electron scattering rate then is derived or inferred from the Drude’s formula via Eq. (6) in the strange metal region. Therefore, if n and m^* are T –independent, the scattering rate \hbar/τ will exhibit a linear–in– T Planckian behavior, i.e. $\hbar/\tau \propto k_B T$. On the contrary, if the effective mass varies with temperatures, the scattering rate would deviate from Planckian, i.e., \hbar/τ is not linear–in– T (but electrical resistivity in this case is still T –linear). Discussions about this potential deviation of scattering rate can be found from Section 4.1 in Ref. 17 of our manuscript [S. A. Hartnoll and A. P. Mackenzie, arXiv:2107.07802].

Now, we address the question concerning the narrow blue temperature window in the region with linear–in– T resistivity (slightly above the Fermi–liquid crossover T_{FL}), schematically shown in Figure 1b of our previous manuscript. We stated that only within this temperature window does the scattering rate exhibit Planckian property, namely $1/\tau \propto T$. Now, we elaborate on the reason why we schematically showed the narrow Planckian region in Figure 1b of our previous manuscript: The authors in arXiv: 2107.07802 argued and showed that in the T –linear region $T_{FL} < T < T^*$, the effective mass may be in general temperature dependent (T^* is the upper cutoff temperature for the T –linear resistivity). Examples are systems with electron–phonon interaction or the effective marginal Fermi liquid. For instance, as shown in arXiv:2107.07802, the renormalization of the effective

mass m^* in marginal Fermi liquid theory takes the following temperature–dependent form:

$$\frac{m^*(T)}{m_0} \propto 1 - \left. \frac{\partial}{\partial \omega} \Sigma'(\omega, T) \right|_{\omega=0} = 1 + \frac{2\lambda}{\pi} \log \frac{T^*}{T}$$

with Σ' being the real part of self–energy, λ being a coupling constant, and m_0 being the bare mass [refer to S. A. Hartnoll and A. P. Mackenzie, arXiv:2107.07802 (2021)]. The scattering rate satisfies the following temperature dependence,

$$\frac{1}{\tau} \propto \frac{\rho}{m^*} \propto \frac{A_1 T}{1 + \frac{2\lambda}{\pi} \log \frac{T^*}{T}}, \quad (\rho \sim A_1 T).$$

Consequently, $1/\tau$ deviates from the Planckian scattering rate in the above case.

On the other hand, it can be shown that the effective mass is independent of temperature in a Fermi–liquid phase (see the proof below and arXiv:2107.07802). By continuity, the temperature regime where m^* is T –independent would extend from low temperature ($T < T_{FL}$) to the temperature regime ($T \sim T_{FL}$) inside the lower end with linear–in– T resistivity. Within this narrow temperature window slightly higher than T_{FL} , the effective mass can still be well described as temperature independent, indicating that the scattering rate is Planckian only in that narrow temperature window, provided that the electron effective mass is T –dependent in the T –linear–resistivity region. This narrow temperature window corresponds to the blue region shown in Figure 1b. The above justification is referred to Section 4.1 and Fig. 6 in Ref. 17 in our former manuscript.

In our previous manuscript, we adapted the argument in Ref. 17 and assumed that electron effective mass in the T –linear resistivity region is also temperature dependent. As a result, the Planckian region is restricted to the narrow strip close to the boarder of Fermi liquid. However, up to the time we submitted our manuscript, we have not carefully checked within our theory whether or not the electron effective mass is indeed temperature dependent. We thank the reviewer to raise this question so that we have a chance to more carefully check our previous assumption. After careful calculations on the electron effective mass in the strange metal T –linear resistivity region of our system, we find that the temperature dependence of electron effective mass is negligible. As a result, our Planckian state extends to the entire strange metal region with T –linear resistivity (instead of restricting to the narrow strip close to T_{FL} as we stated in our previous manuscript). Below we provide the derivation of the nearly temperature independent electron effective mass in our case.

We first provide the derivation of the well–known result that the electron effective mass in a Fermi–liquid phase is temperature independent: Using the following relations

$$\frac{m^*(T)}{m_0} \propto 1 - \left. \frac{\partial}{\partial \omega} \Sigma'(\omega, T) \right|_{\omega=0} \quad (\text{mass renormalization})$$

$$\text{and } \Sigma'(\omega, T) = \frac{1}{\pi} \mathcal{P} \int_{-\infty}^{\infty} \frac{\Sigma''(\omega', T)}{\omega' - \omega} d\omega' \quad (\text{Kronig–Kramer's relation})$$

with $\Sigma'(\Sigma'')$ being the real (imaginary) part of self-energy and m_0 being the bare mass, the correction of the effective mass at finite temperature in the Fermi-liquid regime can be expressed as

$$\begin{aligned} m^*(T) - m_0 &\propto -m_0 \left. \frac{\partial}{\partial \omega} \Sigma'(\omega, T) \right|_{\omega=0} = -\frac{m_0}{\pi} \left[\frac{\partial}{\partial \omega} \mathcal{P} \int_{-\infty}^{\infty} \frac{\Sigma''(\omega', T)}{\omega' - \omega} d\omega' \right] \Big|_{\omega=0} \\ &= -\frac{m_0}{\pi} \mathcal{P} \int_{-\infty}^{\infty} \frac{\Sigma''(\omega', T) - \Sigma''(0, T)}{(\omega')^2} d\omega'. \end{aligned}$$

In the Fermi-liquid phase, the imaginary part of the self-energy takes the form of $\Sigma''(\omega', T) \propto (\pi k_B T)^2 + (\hbar \omega')^2$, indicating that $\Sigma''(\omega', T) - \Sigma''(0, T) \propto (\omega')^2$ is independent of temperature in the above equation of $m^*(T) - m_0$. Hence, the effective mass, $m^*(T)$, at finite temperature in the Fermi-liquid phase is independent of temperature.

Now, we check whether the electron effective mass in the strange metal region of our model system is temperature dependent or not. To this end, we compute the temperature dependence of the effective mass due to the Kondo term in our model based on the equation,

$$\frac{m^*(T)}{m_0} \propto 1 - \left. \frac{\partial}{\partial \omega} \Sigma'_c(\omega, T) \right|_{\omega=0}$$

with $\Sigma'_c(\omega, T)$ being the real part of the conduction-electron self-energy. We find $\Sigma'_c(\omega, T) = C + \frac{2\Lambda T}{\lambda^2} \ln\left(\frac{T x_0}{m_\chi}\right)$ and thus the effective mass shows a $T \ln T$ temperature dependence, given by

$$1 - \left. \frac{\partial \Sigma'_c(\omega)}{\partial \omega} \right|_{\omega=0^+} \approx (1 - C) - \frac{2\Lambda T}{\lambda^2} \ln\left(\frac{T x_0}{m_\chi}\right), \quad \text{where } C \equiv \text{Re} \left[\Lambda \int \frac{d\varepsilon}{(\varepsilon + \lambda + i\Gamma)^2} \right].$$

Here, λ is the Lagrange multiplier introduced to enforce the local constraint of the f fermions, Λ denotes the energy cutoff of the χ boson, $m_\chi \propto J_K - J_K^*$ is the mass (or energy gap) of the χ boson, and $\Gamma = 2\pi J_K^2 N_0$ (N_0 is the conduction-electron density of states at Fermi energy). x_0 is a dimensionless upper cutoff which leads to $1/(e^{x_0} - 1) \approx 1/x_0$. Since the Lagrange multiplier λ has to be sent to infinity $\lambda \rightarrow \infty$ to ensure the local constraint of the f fermions, the temperature dependence of the electron effective mass $m^*(T)/m_0$ can therefore be neglected. We thus conclude that the Planckian state in our system extends over the entire strange-metal region.

R1-B-C

To address the issues of temperature dependence of effective mass and its link to the Planckian scattering rate, we made the following changes in our revised paper:

- We removed the narrow blue region from Figure 1b of the previous manuscript. We also erased the following sentence in the caption of the previous version of Figure

1: “The Planckian scattering rate is reached in the vicinity of T_{FL} (blue area).” The labellings of the Planckian state in Figure 1 were modified as well.

- In the paragraph of the before–concluding remark, we replaced the sentences “Firstly, since m^* is likely temperature dependent in the SM region, Planckian scattering rate is reached only.” with the revised ones “Firstly, in the SM region, the electron effective mass m^* in general may be temperature dependent. In such case, the Planckian scattering rate is reached only.”.
- We added the following sentences in the discussion to elaborate on the temperature dependence of m^* within our theoretical framework: “Nevertheless, we have checked within our theoretical framework that the temperature dependence of m^* in our SM region is negligible. As a result, the Planckian state in our system extends over the entire SM region.”.

R1–C

Can the author’s model capture the B–field dependence of resistivity/specific heat? The current model captures T dependence but I wonder if T/B scaling can emerge from it also.

R1–C–R

In R1–C, Reviewer 1 ask whether or not our model can explain the B –field dependence of the resistivity and the specific heat (coefficient), as well as their T/B scaling. Since we do not explicitly include the B field in our model (the Kondo–Heisenberg model), we do not expect that our model is able to directly capture the B –field dependence of the resistivity and the specific heat (coefficient), as well as their T/B scaling. In our manuscript, in order to confirm that there is indeed a quantum critical point at Nd doping $x \approx 0.03$ hidden beneath the superconducting dome at zero field for $\text{Ce}_{1-x}\text{Nd}_x\text{CoIn}_5$, we observe and present the T/B scaling of the resistivity (Figures 3c and 3d) and the specific heat coefficient (Figures 4c and 4d) for $x = 0.02$ and $x = 0.05$. Though our theoretical model cannot directly capture the field–dependent resistivity and specific heat and their T/B –scaling, we nevertheless argue that the T/B –scaling of observables is expected to happen in our system near a quantum critical point at $x_c \sim 0.03$, in parallel to the ω/T scaling in dynamical observables at criticality. In a quantum–critical system, the energy scale of the long–wavelength quantum–mechanical fluctuations is essential in the determination of the quantum criticality at low temperatures. There exists an emergent scaling of quantum–mechanical energy ($E = \hbar\omega$) and thermal energy ($k_B T$), which is known as the energy–to–temperature (E/T) or frequency–to–temperature (ω/T) scaling. Consider a spin–1/2 particle is subjected to an external (static) magnetic field, this magnetic field sets an magnetic energy scale $E = -\mu_B B$ with μ_B being the Bohr magneton. In analogy with

the well-known ω/T -scaling, a mutual scaling between the magnetic energy and thermal energy is thus expected near the quantum criticality, leading to the T/B -scaling.

R1-C-C

To answer the question mentioned in R1-C, theoretical calculations based on the Kondo lattice model with the inclusion of quantum fluctuations are needed. Since most of the theory part has been removed from the revised manuscript, there is no changes/revisions made in accordance with R1-C.

R1-D

Minor aspects: 1. The phase diagram in Fig. 1 needs some improvements. The $B = 0$ side of the diagram is not clear visually. I suggest adding a panel c to show the projection of the 3D phase diagram onto the $B = 0$ facet. 2. On line 123, the reference to Fig. 3c seems to be problematic.

R1-D-R

We agree with point 1 and point 2 Reviewer 1 mentioned in R1-D. For point 1, the $B = 0$ side of the phases diagram (Figure 1) in our previous manuscript is hard to see clearly. We have added a new figure showing the $B = 0$ projection of the 3D phase diagram in Figure 1. Regarding to point 2, the reference to Fig. 3c on line 123 is a typo. The correct reference is Figs. 3a and 3b.

R1-D-C

In response to the questions in R1-D, we made the following changes in our revised manuscript:

- We added a new figure showing the $B = 0$ projection of the 3D phase diagram in Figure 1 (with labelling Figure 1b). The caption of Figure 1 has also modified in accordance with the newly added figure.
- The typo concerning the figure reference on line 123 has been corrected. The correct reference is “Figs. 3a and 3b”.

Reviewer 2

This paper tried to get the quantum-critical T/B -scaling of the resistivity and heat capacity data of heavy-electron superconductor $Ce_{1-x}Nd_xCoIn_5$. Then the authors provide

a microscopic mechanism to account for the Planckian state in a quantum critical system based on the critical charge fluctuations near Kondo breakdown transition at x_c within the quasi-two-dimensional Kondo–Heisenberg lattice model.

R2–A

I find this paper is not well organized and it is hard for the readers to follow the authors to get a convincing conclusion. I don't think this paper is suitable for Nature Communications.

R2–A–R

In R2–A, Reviewer 2 criticizes on the organization and readability of our paper, rendering our paper not suitable for publication in Nature Communications.

We agree with the reviewer that the original manuscript needs to be revised in a more organized way. In our original manuscript, the experimental results were not put in the most logical fashion. We stated the conclusions of the experimental findings before we provide the evidences. At the end of page 2 we stated “Moreover, clear quantum–critical temperature–to–field–scalings in resistivity and specific heat coefficient in Planckian state are observed near x_c (see below). These observations strongly suggest (i) Kondo hybridization is quantum critical at x_c , and (ii) α depends on the strength of Kondo hybridization in the Planckian SM region, as the second possible scenario mentioned above.” before we provided the experimental evidences. In the revised manuscript, we re–organized the presentation on the experimental results step by step in a more logical way.

On the theory part, in the original manuscript, we provide not only the qualitative understanding of the Planckian state and quantum critical properties, but also intensive and quantitative calculations, including the Planckian α coefficient, and power–law–in– T and power–law–in–doping scaling in specific heat coefficient. Since the theoretical model we applied here has been studied to some extent in Refs. 35, 36, we only summarized the main new results and referred to our previous theoretical papers Refs. 35 and 36 for more technical details. The over–condensing theoretical calculations reduce the readability of our manuscript, especially the details on calculating T –linear resistivity, Planckian coefficient α and scaling of specific heat coefficient. To increase the readability of the paper, in the revised version, we have removed the theoretical analysis, appearing both in the main text and Supplemental Materials (sections S.IV, S.V, and S.VI), of the T –linear resistivity, the specific heat coefficient, and the RG analysis on the QCP based on Refs. 35 and 36 (as is also suggested by Reviewer 3).

To enhance the broad appeal and general readability of our paper, we think it is useful and necessary to provide a qualitative understanding of all the experimental observations within a microscopic model Hamiltonian. Without going to technical calculations in our

revised manuscript, we keep the general theoretical framework—the competition between RVB spin–liquid and Kondo correlations in the Kondo–Heisenberg lattice model used in Refs. 35 and 36 of our former manuscript. This theoretical framework offers a qualitative understanding of the T –linear resistivity, power–law singular in specific heat coefficient and unconventional superconductivity in the “115” family near Kondo breakdown quantum critical point (CeMIn₅ with M = Co, Rh, Ir). We find this theory also applies for our case Ce_{1–x}Nd_xCoIn₅ near $x_c \sim 0.03$ due to its striking similarity in phase diagram and observables to the known results of 115 family. Meanwhile, within the mean–field approach to the Kondo–Heisenberg model, the Kondo coherent scale T_{coh} (PRL 85, 1048 (2000)) and T_c (Ref. 36) depend quadratically on the Kondo hybridization strength. When combining this relation with linear–in–Nd–doping dependence of T_{coh} , $1/\alpha$, T_c , and $1/A_1$ observed experimentally (Fig. 2 of our previous manuscript), we find the Planckian coefficient $1/\alpha$ is quadratically proportional to the Kondo coupling J_K : $\alpha \sim 1/T_{\text{coh}} \sim 1/T_c \sim 1/J_K^2$, one of the main results of this paper.

Following the comments by Reviewers 2 and 3, we have significantly revised and re–organized the manuscript in a much more logical and cleaner way. In the revised manuscript, we emphasize the link among Planckian scattering, quantum critical point and quantum critical scaling, as well as the relation between Planckian coefficient and the Kondo correlation in a more organized manner: We first provide the temperature–field–doping phase diagram where the co–existing superconducting and long–ranged antiferromagnetic order ends at an antiferromagnetic quantum critical point near $x \sim 0.02$. Next, we show the experimental evidence of the Planckian scattering rate in the strange metal region with T –linear resistivity. Then, we present the experimental observations on the linear–in–doping relations of various experimental observables: the coherent Kondo scale T_{coh} , superconducting transition temperature T_c , inverse Planckian coefficient $1/\alpha$, and inverse slope $1/A_1$ of T –linear resistivity. These linear–in–doping relations indicate the inverse relation between α coefficient and Kondo hybridization. To further investigate how α depends on Kondo hybridization near the critical doping x_c and its relation to the possible QCP hidden inside the superconducting dome, we apply an external magnetic field and study the experimental signatures of quantum critical scaling in resistivity and specific heat coefficient. To this end, we provide experimental results showing clear quantum–critical temperature–to–field–scalings in resistivity and specific heat coefficient in Planckian state and the power–law–in– x singular in specific coefficient near $x_c \sim 0.03$. This strongly suggests that Kondo hybridization is quantum critical at x_c .

To qualitatively understand the above strange metal behaviors and more importantly the relation between Planckian coefficient α and Kondo hybridization near the QCP at x_c , we propose a microscopic mechanism based on the large– N Kondo–Heisenberg lattice model studied in Refs. 35 and 36 of our previous manuscript. This theoretical framework offers a qualitative understanding of the link to all the above strange metal phenomena. In particular, since T_{coh} depends quadratically on the Kondo hybridization, $T_{\text{coh}} \sim J_K^2$ [see PRL 85, 1048 (2000)], the linear relations between T_{coh} , $1/A_1$ and $1/\alpha$ suggests the relation

between α and the inverse of quadratic Kondo hybridization strength: $\alpha \sim 1/T_{\text{coh}} \sim 1/J_K^2$.

We believe that the revised manuscript provides clear experimental evidences and qualitative theoretical understanding of the link among Planckian state, Kondo breakdown quantum critical point, which has significantly enhanced the readability of our paper in a broad audience and make our conclusions more convincing.

To highlight the merits and significance of our work, here we would like to point out the positive feedback from Reviewers 1 and 3:

- Reviewer 1: “The analysis clarifies the microscopic mechanism for Planckian scattering observed in this compound. It therefore unifies the physics of a few concepts, including strange metal behavior, Planckian scattering, and AF–KB quantum criticality.”
- Reviewer 3: “This paper contains some very interesting experimental results on strange metallic behavior with T –linear resistivity and Planckian dissipation in Nd–doped CeCoIn_5 . Not only is T –linear resistivity found close to the critical doping, but other signatures of quantum criticality such as strongly interaction–renormalized effective electron mass, and universal magnetic field dependencies of the transport scattering rate are also found. The experimental observations by themselves seem enough to warrant publication of this paper in Nature Communications as far as I am concerned.”

Clearly, both Reviewers 1 and 3 find our results convincing enough “to clarify and unify a few concepts (strange metal, Planckian scattering rate and AF–KB quantum criticality” and “the experimental observations by themselves seem enough to warrant publication of this paper in Nat. Comm.”

Finally, we would like to emphasize again the merits and breakthrough of this work here: The perfect linear–in–temperature resistivity associated with universal linear–in–temperature scattering rate: $1/\tau = \alpha k_B T/\hbar$ with $\alpha \sim O(1)$, so–called the Planckian metal state, has been observed in the normal state of a variety of strongly correlated superconductors close to a quantum critical point. However, the microscopic origin of this intriguing phenomena and its link to quantum criticality still remains an outstanding open problem in condensed matter physics. In this work, we clearly observe the universal quantum–critical temperature–over–field–scaling (T/B –scaling) of the Planckian metal state in the resistivity (with $\alpha \sim O(1)$), and heat capacity of heavy–electron superconductor $\text{Ce}_{1-x}\text{Nd}_x\text{CoIn}_5$ in magnetic fields near the edge of anti–ferromagnetism, driven by critical Kondo hybridization at the critical doping $x_c \sim 0.03$. We also provide strong evidences to support that Kondo hybridization is quantum critical at x_c . We further provide a microscopic mechanism to qualitatively account for the Planckian strange metal state near Kondo breakdown transition at x_c . This mechanism simultaneously captures the T –linear resistivity as well as the quantum–critical scaling and power–law divergence in thermodynamic observables near criticality. More interestingly, we find the Planckian coefficient α is a non–universal

constant which depends inversely on the square of the Kondo hybridization strength. Our mechanism is generic to Planckian metal states in a variety of quantum critical superconductors associated with Fermi surface reconstruction induced by critical charge fluctuations.

Our thorough experimental and theoretical study on this outstanding open problem offers the first comprehensive understanding of the Planckian metal state in any quantum critical systems. We believe that our work constitutes a major breakthrough in the field of correlated electron systems and shows a broad appeal to both theoretical and experimental physics communities. By extension, we feel that our manuscript is appropriate to be published in Nature Communications.

R2–A–C

To address the issues of the organization and readability of our paper raised by Reviewer 2, we only kept the general theoretical framework and removed most of the theoretical analysis, appears both in the main text and Supplemental Materials (sections S.IV, S.V, and S.VI). Meanwhile, we instead presented the theoretical explanations for the linear–in–doping relations of various experimental observables T_{coh} , T_c , $1/\alpha$, and inverse slope $1/A_1$ of T –linear resistivity, which were originally shown in section S.II in the previous version of Supplemental Materials. To this end, we made the following changes in both the revised main text and Supplemental Materials:

- The sentences below “Moreover, clear quantum–critical temperature–to–field–scalings in resistivity and specific heat coefficient in Planckian state are observed near x_c (see below). These observations strongly suggest (i) Kondo hybridization is quantum critical at x_c , and (ii) α depends on the strength of Kondo hybridization in the Planckian SM region, as the second possible scenario mentioned above.” were removed to re–organize the presentation of the experimental results in a more logical fashion.
- The sentence “This mechanism simultaneously captures the observed universal Planckian scattering rate as well as the quantum–critical scaling and power–law divergence in thermodynamic observables near criticality. Our mechanism is generic to Planckian metal states in a variety of quantum critical superconductors near Kondo destruction.” was removed from the abstract.
- We also deleted this sentence “Meanwhile, the critical Kondo coupling therein depends on the anomalous dimension η of our quasi–2d model (see below).” from the manuscript since it contains some details concerning the fractional dimension (η) of the theoretical model in Refs. 35 and 36 in the previous version.
- We removed the details of the theoretical analysis including the sentences and equations of the T –linear resistivity, the specific heat coefficient, and the RG analysis

on the QCP based on Refs. 35 and 36, appearing in the previous version of main text. Meanwhile, to make our paper less technical, we only retained the Kondo lattice Hamiltonian and the general forms of the decomposed Kondo and RKKY terms [Eqs. (4) and (5) of the previous manuscript]. Consequently, the Kondo and RKKY Hamiltonians at and beyond mean-field level were removed [Eqs. (6) and (7) of the previous manuscript]. We also erased the whole sections, containing the detailed calculations of the RG analysis, the T -linear resistivity, and the specific heat coefficient (sections S.IV, S.V and S.VI), in the previous Supplemental Materials.

- The theoretical fittings of C/T in Figures 4a, 4b and S1a based on Eq. (3) in our previous manuscript have been removed.
- We added the sentence “To qualitatively understand the above strange metal behaviors and more importantly the relation between Planckian coefficient α and Kondo hybridization near the QCP at x_c , here, we propose.”
- To qualitatively summarize our theoretical framework, we replaced the sentence “Our proposed mechanism based on. reaches the universal Planckian limit.” with “We propose a microscopic mechanism based on the quasi-2d Kondo-Heisenberg lattice model to qualitatively account for the observed strange metal behaviors.” in the revised summary.
- Regarding to the changes in the theoretical explanations on the linear-in-doping relations of T_{coh} , T_c , $1/\alpha$, and $1/A_1$, we made the following changes:
 - We removed section S.II in the previous version of Supplemental Materials completely.
 - We added the following sentences in the revised abstract: “Meanwhile, we observe the power-law in $|x - x_c|$ singularity of the low-temperature specific heat coefficient, and the perfect linear relations among Kondo coherence scale, inverse of slope in T -linear resistivity and $1/\alpha$ as a function of doping. These constitute clear experimental evidences of Kondo hybridization being quantum critical at x_c .” and “Our study suggests that α is a non-universal constant and depends inversely on the square of Kondo hybridization strength.”
 - The sentence “Eq. (1) above suggests that the Planckian coefficient α inversely depends on the strength of Kondo hybridization in the Planckian SM region, as the second possible scenario mentioned above.” was added in the main text.
 - To show the theoretical explanations for the linear-in-doping of various observables, we included following sentences in the revised main text: “We now apply the above theoretical framework. that $\alpha \propto 1/J_K^2$ consistent with Eq. (2) above.”

- We revised Eqs. (S1)–(S3), originally shown in the section S.II of the previous Supplemental Materials. The revised equations were presented in the Eqs (1)–(3) of the revised manuscript:

$$\text{Eq. (1): } T_{\text{coh}}(x) \propto 1/A_1(x) \propto 1/\alpha(x)$$

$$\text{Eq. (2): } \alpha(x) \propto A_1(x) \propto 1/T_{\text{coh}}(x) \propto 1/J_K^2$$

$$\text{Eq. (3): } T_c \propto \chi^2 \Delta_{RVB} \propto J_K^2 \Delta_{RVB}$$

- In the revised summary, we included the sentence “Furthermore, we find that the Planckian coefficient α for Nd-doped CeCoIn_5 shows an inversely quadratic dependence in the Kondo coupling, as inferred from our theoretical framework as well as the linear relation between $1/\alpha$, the inverse slope of T-linear resistivity $1/A_1$ and the coherence temperature T_{coh} .”

R2-B

Some minor comments: (1) In the caption of Table 1, $\text{Nd}_x\text{Ce}_{1-x}\text{CoIn}_5$ should be $\text{Ce}_{1-x}\text{Nd}_x\text{CoIn}_5$, in order to be consistent with other parts of this manuscript. (2) The format of chemical formula in many references is not correct. For examples, $\text{baf}_2(\text{As}_{1-x}\text{P}_x)_2$ in Ref. 6, $\text{Fes}_{1-x}\text{s}_x$ in Ref. 8, cecoin_5 in Ref. 27, 38, 40.

R2-B-R

We agree with Reviewer 2 that there are some inconsistencies and improper expressions in the format of chemical formula in our previous manuscript. These minor errors Reviewer 2 pointed out have been corrected in our revised manuscript.

R2-B-C

The inconsistent notations/chemical formula in the caption of Table 1 and in the references have been corrected.

Reviewer 3

This paper contains some very interesting experimental results on strange metallic behavior with T-linear resistivity and Planckian dissipation in Nd-doped CeCoIn_5 . Not only is T-linear resistivity found close to the critical doping, but other signatures of quantum criticality such as strongly interaction-renormalized effective electron mass, and universal

magnetic field dependencies of the transport scattering rate are also found. The experimental observations by themselves seem enough to warrant publication of this paper in Nature Communications as far as I am concerned.

However, the theoretical analysis is hard to follow and not clearly presented, and I am therefore not able to determine its validity. I feel that it would be more appropriate to move the theoretical analysis to a separate paper in a specialized journal. I also am confused about certain steps in the theoretical analysis as detailed below:

R3–A

How is the fermion self energy Σ_c , used in section S. V, computed? It would be useful to show the derivation of this. I do not understand how Eq. (S34) is derived!

R3–A–R

In R3–A, Reviewer 3 pointed out that there was no derivation of the conduction–electron self–energy Σ_c in Eq. (S34) of our manuscript. Below, we summarize the derivations of Σ_c by including the key equations.

By Wick theorem, the self–energy Σ_c can be derived as the following:

$$\begin{aligned}\Sigma_c(i\omega, \mathbf{k}) &= \left(\frac{J_K^2}{\beta}\right) \sum_{\mathbf{p}} \sum_{\nu} G^f(\mathbf{p}, \nu) G^x(\mathbf{k} - \mathbf{p}, \omega - \nu) \\ &= \left(\frac{J_K^2}{\zeta\beta}\right) \sum_{\mathbf{p}} \sum_{\nu} \frac{1}{i\nu - \lambda - \Sigma_f^{(2)}(\nu)} \cdot \left[\frac{1}{i\nu - i\omega - \zeta^{-1}\varepsilon_x(\mathbf{k} - \mathbf{p})} - \frac{1}{i\nu - i\omega + \zeta^{-1}\varepsilon_x(\mathbf{k} - \mathbf{p})} \right]\end{aligned}$$

where G^f (G^x) denotes the Green’s functions of the local f fermion (χ boson) and $\zeta \equiv \frac{J_K^2 N_0}{\Lambda}$ is a constant. Here, $\Sigma_f^{(2)}(\nu) = i \operatorname{sgn}(\nu) \Gamma$ with $\Gamma = 2\pi J_K^2 N_0$. Here, $\varepsilon_x(\mathbf{k}) = J_K + \frac{J_K^2 N_0 \lambda}{\Lambda} - \frac{J_K^2 N_0}{\Lambda} \varepsilon(\mathbf{k})$ denotes the dispersion of the χ boson, see R3–C below for detailed discussion/derivations of ε_x . Due to the sign function, we should deal with the Matsubara sum over $\nu > 0$ and $\nu < 0$, separately and express $\Sigma_c = \Sigma_c^> + \Sigma_c^<$ with

$$\Sigma_c^{\geq}(i\omega, \mathbf{k}) = \left(-\frac{J_K^2}{\beta}\right) \sum_{\mathbf{p}} \sum_{\nu \geq 0} G^f(\mathbf{p}, \nu) G^x(\mathbf{k} - \mathbf{p}, \omega - \nu).$$

After performing the Matsubara sum over the upper plane of the Matsubara frequency, we obtain

$$\begin{aligned}\Sigma_c^>(i\omega, \mathbf{k}) &= \left(-\frac{J_K^2}{\zeta}\right) \sum_{\mathbf{p}} \frac{1}{i\omega - \zeta^{-1}\varepsilon_x(\mathbf{k} - \mathbf{p}) - \lambda - i\Gamma}, & \text{for } \omega > 0, \\ \Sigma_c^<(i\omega, \mathbf{k}) &= \left(-\frac{J_K^2}{\zeta}\right) \sum_{\mathbf{p}} \frac{1}{i\omega - \zeta^{-1}\varepsilon_x(\mathbf{k} - \mathbf{p}) - \lambda + i\Gamma}, & \text{for } \omega > 0.\end{aligned}$$

Using $\sum_{\mathbf{p}} \rightarrow N_0 \int_0^\Lambda d\varepsilon$ and taking the imaginary part of $\Sigma_c^>(i\omega \rightarrow \omega + i\epsilon, \mathbf{k}) \Big|_{\epsilon=0^+}$, we have

$$\begin{aligned} \text{Im}\Sigma_c^>(\omega > 0) &= \left(-\frac{J_K^2}{\zeta}\right) \text{Im} \int_0^\Lambda \frac{N_0 d\varepsilon}{\omega - \varepsilon - i\Gamma} \\ &= \left(\frac{J_K^2 N_0}{\zeta}\right) \left[\tan^{-1}\left(\frac{\Lambda - \omega}{\Gamma}\right) + \tan^{-1}\left(\frac{\omega}{\Gamma}\right) \right] \\ (\omega/\Lambda \ll 1) &\approx \left(\frac{J_K^2 N_0}{\zeta}\right) \left[\tan^{-1}\left(\frac{\Lambda}{\Gamma}\right) + \tan^{-1}\left(\frac{\omega}{\Gamma}\right) \right]. \end{aligned}$$

If we further assume that $\Lambda/\Gamma \gg 1$ and $\omega/\Gamma \ll 1$, we therefore have $\tan^{-1}(\Lambda/\Gamma) \approx \pi/2$ and $\tan^{-1}(\omega/\Gamma) \approx \omega/\Gamma$. Finally, we find that the approximated imaginary part of conduction-electron self-energy $\text{Im}\Sigma_c^>$ to the first order in ω , given by

$$\text{Im}\Sigma_c^>(\omega > 0) \approx \left(\frac{J_K^2 N_0}{\zeta}\right) \left[\frac{\pi}{2} + \frac{\omega}{\Gamma}\right] \equiv (\Omega + \varsigma\omega)$$

$$\text{with } \Omega \equiv \frac{\Lambda\pi}{2}, \quad \varsigma \equiv \frac{\Lambda}{\Gamma} = \frac{\Lambda}{2\pi J_K^2 N_0} = \frac{1}{2\pi J_K^2 N_0^2} \equiv \frac{1}{2\pi j_K^2}.$$

with $j_K \equiv J_K N_0$. Following a similar approach, we have

$$\text{Im}\Sigma_c^<(\omega < 0) = -(\Omega + \varsigma\omega).$$

We thus obtain Eq. (S34), given by

$$\text{Im}\Sigma_c(\omega) = \text{sgn}(\omega)(\Omega + \varsigma\omega).$$

R3-A-C

Since the detail of the calculations of the electrical resistivity has been removed from the revised manuscript, there is no changes/revisions made in accordance with R3-A.

R3-B

Why is the fermion self energy being taken to be equal to the transport scattering rate (Eq. (S33))? What about vertex corrections contributing to transport? If the self-energy represents largely forward-scattering processes at the QCP, then it is NOT equal to the transport scattering rate. Again, whether this is true or not is not clear because of the highly compressed theoretical section of the paper.

R3–B–R

We thank Reviewer 3 to point out the important issue of scattering rate.

We agree with Reviewer 3 that the transport scattering rate (inverse of transport time) as Reviewer 3 mentioned, in general, differs from the (imaginary part of) the conduction electron self–energy [see Fig. 1(a)] and the transport scattering rate is appropriate for the calculation of resistivity. Thus, their difference deserves further discussions: The relaxation time (or lifetime) $\tau_{\text{life}}(\mathbf{k})$ of a particular state \mathbf{k} is defined as the inverse of the total scattering rate of \mathbf{k} , given by

$$\frac{1}{\tau_{\text{life}}(\mathbf{k}, \omega)} = \sum_{\mathbf{k}'} W_{\mathbf{k}'\mathbf{k}} = -2\text{Im}\Sigma_c(\mathbf{k}, \omega) \quad (7)$$

with $W_{\mathbf{k}\mathbf{k}'}$ being the scattering rate for the transition $\mathbf{k} \rightarrow \mathbf{k}'$. The relaxation rate $1/\tau_{\text{life}}(\mathbf{k}, \omega)$ above measures the rate a given state \mathbf{k} being scattered out. The contribution of self–energy largely represents the forward scattering as stated by Reviewer 3. From the Boltzmann theory, however, the transport time $\tau_{\text{tr}}(\mathbf{k})$ of the \mathbf{k} state for a “single–band” system with only one type of fermions or conduction electrons can be evaluated as

$$\frac{1}{\tau_{\text{tr}}(\mathbf{k}, \omega)} \propto \sum_{\mathbf{k}'} W_{\mathbf{k}'\mathbf{k}} (n_{\mathbf{k}} - n_{\mathbf{k}'}) \quad (8)$$

with $n_{\mathbf{k}}, n_{\mathbf{k}'}$ being the non–equilibrium distribution functions of the initial and final states. Evidently, $\tau_{\text{life}}(\mathbf{k}, \omega)$ differs from $\tau_{\text{tr}}(\mathbf{k}, \omega)$ by the factor $n_{\mathbf{k}} - n_{\mathbf{k}'}$ which takes into account the information of particle filling of the initial and final states. This factor implies that, if the state \mathbf{k} does not significantly change by the scattering, i.e. the forward scattering $\mathbf{k} \approx \mathbf{k}'$, this small–angle transition makes only little contribution to scattering rate or resistivity since $n_{\mathbf{k}} - n_{\mathbf{k}'} \rightarrow 0$. On the contrary, when there is a large momentum change, namely the backward scattering, the large–angle scattering contributes more significantly to the scattering rate or resistivity as the large–angle scattering is more effective on dissipating electrical current. For a particular case—the elastic impurity scattering in isotropic materials, the transport scattering rate of Eq. (8) can be reduced to a simpler form,

$$\frac{1}{\tau_{\text{tr}}(\mathbf{k}, \omega)} = \frac{1}{V} \sum_{\mathbf{k}'} W_{\mathbf{k}'\mathbf{k}} (1 - \cos \theta_{\mathbf{k}\mathbf{k}'}) \quad (9)$$

with $\theta_{\mathbf{k}\mathbf{k}'}$ being the angle between \mathbf{k} and \mathbf{k}' . The $1 - \cos \theta_{\mathbf{k}\mathbf{k}'}$ factor here plays the same role with $n_{\mathbf{k}} - n_{\mathbf{k}'}$ in Eq. (8). It is worthwhile of noticing that Eq. (8) and its associated assertions are only applicable for the intra–band transition since the non–equilibrium distribution functions of the initial and final states therein belong to the same band.

In our work, we indeed took the lifetime (inverse of the relaxation rate) of the conduction electron as the transport scattering time in the calculation of the electronic resistivity through the Boltzmann transport theory, i.e.

$$\rho^{-1}(T) = \left(-\frac{ne^2}{m^*} \right) \int \tau_{\text{life}}(\omega) \frac{\partial f(\omega)}{\partial \omega} d\omega$$

Figure 1: Feynman diagrams for (a) the leading non-trivial self energy of conduction electron Σ_c and (b) the vertex correction U_K .

with $f(\omega) \equiv (e^{\beta\omega} + 1)^{-1}$ being the Fermi function.

At first sight, it is tempting to think that our approach would be problematic as we associated the lifetime ($\tau_{\text{lif}}^{-1}(\mathbf{k}, \omega) = -2\text{Im}\Sigma_c(\mathbf{k}, \omega)$) of conduction electron with the transport scattering time, as the self-energy (or forward scattering) is ineffective on dissipating currents and thus only makes a tiny contribution to resistivity due to the $n_{\mathbf{k}} - n_{\mathbf{k}'}$ factor in Eq. (8). However, we would like to argue that this assertion cannot be simply applied in our case since, in our case, two bands participate in the scattering process simultaneously: an itinerant conduction c -band and a heavy, charge-neutral f -band (also known as spinon). The Kondo term H_K plays the role of scattering a conduction-band electron into the charge-neutral f band or vice versa. The transport scattering rate of this inter-band transition cannot be directly evaluated by applying Eq. (8) which is only suitable for the intra-band (or only one electron band) scattering. Thus, the former analysis on the contribution from scattering of the different angles to the electrical resistivity based on Eq. (8) is not valid in the presence of inter-band scattering. Here, we will qualitatively elaborate on this assertion without invoking technical derivations: Distinct from the relaxation rate $1/\tau_{\text{lif}}$ shown in Eq. (7), the transport scattering rate $1/\tau_{\text{tr}}$ not only contains transition rate W but also needs to include the filling of the initial and final states, as we mentioned the transport time for the intra-band scattering above. In our case, when a conduction-band electron is scattered into the charge-neutral f band by the Kondo term, transition rate of this $c \rightarrow f$ process is described by $W_{\mathbf{k}\mathbf{k}'}(c \rightarrow f) \sim |\langle f_{\mathbf{k}'} | H_K | c_{\mathbf{k}} \rangle|^2$ where the initial (c -electron) and final (f -spinon) states belong to two distinct types of fermion baths. The crucial factor for our arguments comes from the effect of particle filling of the initial and final states. Recall our discussion of the intra-band transition, it is the factor $n_{\mathbf{k}} - n_{\mathbf{k}'}$ or $1 - \cos\theta_{\mathbf{k}\mathbf{k}'}$ in the transport scattering rate that leads to the angular dependence of dissipation on the electric current. If the state \mathbf{k} does not significantly change by the scattering, i.e. the forward scattering $\mathbf{k} \approx \mathbf{k}'$, this small-angle scattering process only makes little contribution to scattering rate or resistivity since $n_{\mathbf{k}} - n_{\mathbf{k}'} \rightarrow 0$. However, unlike the intra-band transition, the initial and final states due to the (inter-band transition) Kondo scattering happens at different bands and thus the transport scattering rate contains the distribution functions of both the conduction and heavy f bands, denoted as $n_{\mathbf{k}}^c$ and $n_{\mathbf{k}'}^f$. In such case, the distribution

functions themselves which account for the dissipation of conduction–band current are not likely to cancel out with each other even when the initial and final states carry the same wave vectors. Hence, we conclude that, in such systems, the forward scattering (or self–energy) can still significantly contribute to the electrical resistivity. Consequently, the f band effectively plays the role of dissipating the conduction electron current during the small–angle processes. In the paper PRL 98, **026402** (2007) by I. Paul et al., the authors considered a two–band (a conduction c –band and a local fermionic spinon f –band) Kondo lattice model in close resemblance to ours, and the transport time τ_{tr} was identified with the lifetime of conduction electron while calculating the electrical resistivity in their work. Therein, an argument similar to ours was also used to justify the validity in their calculation of resistivity. The authors in that PRL paper explicitly stated that “However, our model consists of two bands, one of light particles (the conduction electrons) which scatter from very heavy particles (the spinons) [20]. As such, the charge neutral spinons act as an effective bath for the relaxation of the conduction electron current”. We thus conclude that conduction–electron self–energy in our case can involve in the dissipation of electric current in an efficient way and can thus be applied in the calculation of electrical resistivity.

To address the question concerning the contribution of vertex corrections to the transport, we estimate the real part of the leading non–trivial vertex correction U_K of a certain spin σ to the Kondo term H_K , given by $U_K(k, p, \sigma) \equiv \langle c_{k\sigma}^\dagger f_{p\sigma} \chi_{k-p} \rangle \sim G_\chi G_c G_f$, as shown in Fig. 1(b). Here, k and p denote the four–vector notations $k = (\mathbf{k}, k_n)$ and $p = (\mathbf{p}, p_m)$ with k_n and p_m being the Matsubara frequencies for c and f operators, respectively. While setting the exter

$$\text{Re}[U_K(\omega, \mathbf{k} = 0, \sigma)] = (-2J_K^2 N_0) \left[\frac{1}{\omega - m_\chi} \left(\ln \frac{\lambda}{D + \lambda} - \ln \frac{\omega + D - \lambda}{\omega - D - \lambda} \right) \right].$$

Here, λ denotes the Lagrange multiplier to enforce the local constraint of the f fermions, D is the half–bandwidth of conduction band, and $m_\chi \sim J_K - J_K^*$ effectively represents the mass of the bosonic hybridization fluctuation field χ which is proportional to the distance of the Kondo coupling J_K to its critical value J_K^* . Since the Lagrange multiplier has to be sent to infinity $\lambda \rightarrow \infty$ to ensure the local constraint of the f fermions, we find $\text{Re}[U_K]$ shown above is negligibly small.

R3–B–C

Since the detail of the calculations of the electrical resistivity has been removed from the revised manuscript, there is no changes/revisions made in accordance with R3–B.

R3–C

Similarly, I do not understand the computations of section S. VI. What is the value of $\varepsilon_\chi(k)$? The derivation of this from the χ self energy should be provided! For these reasons, I

believe that the experimental part of the paper is very interesting and publishable, while the theoretical part should be moved to a separate paper in a specialized journal, and written up in a manner that provides systematic and clear derivations of all the necessary steps. If the authors can do this, I can recommend publication of their (experimental) work in Nature Communications.

R3–C–R

We thank Reviewer 3’s comments/suggestions, especially the encouraging comment “the experimental part of the paper is very interesting and publishable” and Reviewer 3 will be willing to recommend publication of our experimental work in Nature Communications. Reviewer 3 also pointed out the lack of details of the theoretical aspect in our previous manuscript: many computations such as the derivation of the self-energy of the χ boson in Supplemental Materials section S.VI are not provided. Thus, Reviewer 3 suggested us to move the theoretical part to a separate paper in a specialized journal.

We accept Reviewer 3’s suggestion to remove the technical parts of our theory from the manuscript, including the calculation of the Planckian coefficient and the singular T -power-law behavior of the specific heat coefficient near x_c .

Nevertheless, to enhance the broad appeal and general readability of our paper, we think it is useful and necessary to provide a qualitative understanding of all the experimental observations within a microscopic model Hamiltonian. Without going to technical calculations, we keep the general theoretical framework—the competition between RVB spin-liquid and Kondo correlations in the Kondo–Heisenberg lattice model used in Refs. 35 and 36 of our previous manuscript—in our revised manuscript. This theoretical framework offers a qualitative understanding of the T -linear resistivity, the singular power-law behavior of specific heat coefficient and unconventional superconductivity in the “115” family (CeMIn₅ with M = Co, Rh, Ir) near Kondo breakdown quantum critical point. We find this theory also applies for our case Ce_{1-x}Nd_xCoIn₅ near $x_c \sim 0.03$ due to its striking similarity in phase diagram and observables to the known results of 115 family. Meanwhile, within the mean-field approach to the Kondo–Heisenberg model, the Kondo coherent scale T_{coh} (PRL85, 1048 (2000)) and T_c (Ref. 36 of our previous manuscript) depend quadratically on the Kondo hybridization strength. When combining this relation with linear-in-Nd-doping dependence of T_{coh} , $1/\alpha$, T_c , and $1/A_1$ observed experimentally, we find the Planckian coefficient $1/\alpha$ is quadratically proportional to the Kondo coupling J_K : $\alpha \sim 1/T_{\text{coh}} \sim 1/T_c \sim 1/J_K^2$, one of the main results of this paper.

Following the comments by Reviewers 2 and 3, we have significantly revised and re-organized the manuscript in a much more logical and cleaner way. In the revised manuscript, we emphasize the link among Planckian scattering, quantum critical point and quantum critical scaling, as well as the relation between Planckian coefficient and the Kondo correlation in a more organized manner: We first provide the temperature–field–doping phase

diagram where the co-existing superconducting and long-ranged antiferromagnetic order ends at an anti-ferromagnetic quantum critical point near $x \sim 0.02$. Next, we show the existence of the Planckian scattering rate in the strange metal region with T -linear resistivity. Then we present the experimental observations on the linear-in-doping relations of various experimental observables: the coherent Kondo scale T_{coh} , superconducting transition temperature T_c , inverse α coefficient, and inverse slope $1/A_1$ of T -linear resistivity. These linear-in-doping relations indicate the inverse relation between α coefficient and Kondo hybridization. To further investigate how α depends on Kondo hybridization near the critical doping x_c and its relation to the possible QCP hidden inside the superconducting dome, we apply an external magnetic field and study signatures of quantum critical scaling in resistivity and specific heat coefficient. Then we provide experimental results showing clear quantum-critical temperature-to-field-scalings in resistivity and specific heat coefficient in Planckian state and the power-law-in- x singularity in specific coefficient near $x_c \sim 0.03$. This strongly suggests that Kondo hybridization is quantum critical at x_c .

To qualitatively understand the above strange metal behaviors and more importantly the relation between Planckian coefficient α and Kondo hybridization near QCP at x_c , we propose a microscopic mechanism based on the large- N Kondo-Heisenberg lattice model studied in Refs. 35 and 36 of our previous draft. This theoretical framework offers a qualitative understanding of the link to all the above phenomena (see paragraph above).

Regarding the derivation of the dispersion ε_χ and the self-energy of the $\hat{\chi}$ boson in section S.VI of the Supplemental Materials that Reviewer 3 mentioned, we provide the detailed derivations below:

At the bare level, the $\hat{\chi}$ boson does not have dynamics and dispersion. Its bare Green's function is given by $G_\chi(\mathbf{q}, \omega) = 1/(-J_K)$. To generate its dynamics and dispersion, we calculate its self-energy to the one-loop order and use RPA theory. The non-trivial contribution comes from the second-order perturbation by the Kondo term H_K . Here, we assume $\hat{\chi}$ to be a real boson (like phonon), it thus obeys $\hat{\chi}_\mathbf{q}^\dagger = \hat{\chi}_{-\mathbf{q}}$ and $\chi_\mathbf{q} = b_\mathbf{q} + b_{-\mathbf{q}}^\dagger$ with $b(b^\dagger)$ being an annihilation (creation) operator. Now, the Green's function of the real $\hat{\chi}$ field can be described as

$$\begin{aligned} G_\chi(\mathbf{q}, \tau) &\equiv -\langle T_\tau \chi_\mathbf{q}(\tau) \chi_{-\mathbf{q}}(0) \rangle = -\langle T_\tau [b_\mathbf{q} + b_{-\mathbf{q}}^\dagger](\tau) [b_{-\mathbf{q}} + b_\mathbf{q}^\dagger](0) \rangle \\ &= -\langle T_\tau b_\mathbf{q}(\tau) b_\mathbf{q}^\dagger(0) \rangle - \langle T_\tau b_{-\mathbf{q}}^\dagger(\tau) b_{-\mathbf{q}}(0) \rangle. \end{aligned}$$

In the frequency domain, the Green's functions of $\hat{\chi}$ shares the same form with that of phonon, $G_\chi(\mathbf{q}, i\omega) = G_b(\mathbf{q}, i\omega) + G_b^*(\mathbf{q}, i\omega)$ where $G_b(\mathbf{q}, i\omega) \propto (i\omega - \varepsilon_\chi(\mathbf{q}))^{-1}$ with $\varepsilon_\chi(\mathbf{q})$ being the dispersion generated by the one-loop self-energy of $\hat{\chi}$ (or the b operator).

The Kondo term takes the following form

$$\begin{aligned}
H_K &= J_K \sum_{\mathbf{q}, \mathbf{k}} \sum_{\sigma} \left[\left(c_{\mathbf{k}\sigma}^{\dagger} f_{\mathbf{q}\sigma} \right) \hat{\chi}_{\mathbf{k}-\mathbf{q}} + H.c. \right] \\
&= J_K \sum_{\mathbf{q}, \mathbf{k}} \sum_{\sigma} \left[\left(c_{\mathbf{k}\sigma}^{\dagger} f_{\mathbf{q}\sigma} \right) \left(b_{\mathbf{k}-\mathbf{q}} + b_{-\mathbf{k}-\mathbf{q}}^{\dagger} \right) + H.c. \right]
\end{aligned}$$

similar to the Yukawa interaction. Once we consider the self-energy of b , we have the Dyson equations of G_b , described as

$$G_b^{-1} = -J_K - \Sigma_b, \quad (G_b^{-1})^* = -J_K - \Sigma_b^*.$$

By Wick theorem, we obtain Σ_b as

$$\begin{aligned}
\Sigma_b(\mathbf{k}, i\omega) &= \left(\frac{J_K^2}{\beta} \right) \sum_{\nu, \mathbf{p}} G_c(\mathbf{p}, i\nu) G_f(\mathbf{p} + \mathbf{k}, i\nu + i\omega) = \left(\frac{J_K^2}{\beta} \right) \sum_{\nu, \mathbf{p}} \frac{1}{i\nu - \varepsilon_{\mathbf{p}}} \frac{1}{i\nu + i\omega - \lambda} \\
&= J_K^2 \sum_{\mathbf{p}} \left(\frac{n_F(\varepsilon_{\mathbf{p}})}{i\omega - \lambda + \varepsilon_{\mathbf{p}}} + \frac{n_F(\lambda)}{-i\omega + \lambda - \varepsilon_{\mathbf{p}}} \right) \approx J_K^2 \sum_{\mathbf{p}, \varepsilon_{\mathbf{p}} \leq 0} \left(\frac{1}{i\omega - \lambda + \varepsilon_{\mathbf{p}}} \right) \\
&= J_K^2 N_0 \int_{\varepsilon < 0} \left(\frac{d\varepsilon}{i\omega - \lambda + \varepsilon} \right)
\end{aligned}$$

To obtain the dispersion of $\hat{\chi}$, we only integrate out the higher frequency modes for which $-\Lambda \leq \varepsilon \leq -\Lambda + \varepsilon_{\mathbf{k}}$ in the energy integral, i.e.

$$\begin{aligned}
\Sigma_b(\mathbf{k}, i\omega) &= J_K^2 \int_{-\Lambda}^{-\Lambda + \varepsilon_{\mathbf{k}}} \left(\frac{1}{i\omega - \lambda + \varepsilon} \right) = J_K^2 N_0 \ln \left[\frac{i\omega - \lambda - \Lambda + \varepsilon_{\mathbf{k}}}{i\omega - \lambda - \Lambda} \right] \\
(\Lambda \gg \omega, \lambda) &\approx J_{\chi}^2 N_0 \ln \left[1 - \frac{i\omega - \lambda + \varepsilon_{\mathbf{k}}}{\Lambda} \right] \\
&\approx -\frac{J_{\chi}^2 N_0}{\Lambda} (i\omega - \lambda + \varepsilon_{\mathbf{k}})
\end{aligned}$$

Thus, once $\Sigma_b(\mathbf{k}, i\omega)$ is obtained above, we have the $\hat{\chi}$ -field Green's function, given by

$$\begin{aligned}
G_{\chi}(\mathbf{k}, i\omega) &= \frac{1}{-J_K^{-1} - \Sigma_b(\mathbf{k}, i\omega)} + \frac{1}{-J_K^{-1} - \Sigma_b^*(\mathbf{k}, i\omega)} \\
&\equiv \frac{1}{i\omega_{\chi} - \varepsilon_{\chi}(\mathbf{k})} - \frac{1}{i\omega_{\chi} + \varepsilon_{\chi}(\mathbf{k})},
\end{aligned}$$

where

$$\omega_{\chi} = \frac{J_{\chi}^2 N_0}{\Lambda} \omega \equiv \zeta \omega, \quad \varepsilon_{\chi}(\mathbf{k}) = J_{\chi} + \zeta \lambda - \zeta \varepsilon_{\mathbf{k}}$$

with $\zeta \equiv \frac{J_{\chi}^2 N_0}{\Lambda}$ being a constant.

R3-C-C

Regarding to the revision of our paper suggested Reviewer 3, we only kept the general theoretical framework and removed most of the theoretical analysis, appears both in the main text and Supplemental Materials (sections S.IV, S.V, and S.VI). Meanwhile, we instead presented the theoretical explanations for the linear-in-doping relations of various experimental observables T_{coh} , T_c , $1/\alpha$, and inverse slope $1/A_1$ of T -linear resistivity, which were originally shown in section S.II in the previous version of Supplemental Materials. The corresponding changes we made in the revised manuscript were shown in R2-A-C.

REVIEWERS' COMMENTS

Reviewer #1 (Remarks to the Author):

The authors have fully addressed my comments so I recommend publication of this manuscript.

Reviewer #2 (Remarks to the Author):

I have carefully read the rebuttal letter and revised manuscript. The manuscript has been significantly revised and its organization is fine now. The physics in current paper is important. I recommend the publication of this paper in Nature Communications.